# Opposing gene regulatory programs governing myofiber development and maturation revealed at single nucleus resolution

Matthieu Dos Santos[1], Akansha M. Shah[1], Yichi Zhang [1],
Svetlana Bezprozvannaya[1], Kenian Chen [2], Lin Xu [2], Weichun Lin[3],
John R. McAnally[1], Rhonda Bassel-Duby [1], Ning Liu [1] & Eric N. Olson [1] ✉

Skeletal muscle fibers express distinct gene programs during development and maturation, but the underlying gene regulatory networks that confer stage-specific myofiber properties remain unknown. To decipher these distinctive gene programs and how they respond to neural activity, we generated a combined multi-omic single-nucleus RNA-seq and ATAC-seq atlas of mouse skeletal muscle development at multiple stages of embryonic, fetal, and postnatal life. We found that Myogenin, Klf5, and Tead4 form a transcriptional complex that synergistically activates the expression of muscle genes in developing myofibers. During myofiber maturation, the transcription factor Maf acts as a transcriptional switch to activate the mature fast muscle gene program. In skeletal muscles of mutant mice lacking voltage-gated L-type $Ca^{2+}$ channels (Cav1.1), Maf expression and myofiber maturation are impaired. These findings provide a transcriptional atlas of muscle development and reveal genetic links between myofiber formation, maturation, and contraction.

Skeletal muscle is the largest tissue in the body, and is essential for locomotion, posture, body temperature regulation, and glucose and amino acid storage. Skeletal muscle is mainly composed of myofibers, which are large contractile cell syncytia containing tens to thousands of nuclei. Myofibers are formed by differentiation of muscle stem cells (MuSCs) into myoblasts, which undergo fusion to form multinucleated myofibers[1]. Like hematopoiesis, myogenesis occurs in successive phases, starting from formation of primary myofibers during embryogenesis to secondary myofibers during the fetal period. After birth, primary and secondary myofibers mature and grow extensively by synthesizing new proteins and fusing with myoblasts to reach their mature adult size[2].

Numerous transcription factors (TFs) have been shown to regulate myogenesis. The paired-homeobox factors Pax3 and Pax7 determine the fate of MuSCs[3], while the myogenic regulatory factors, Myf5, MyoD, Myogenin, and Mrf4, promote the differentiation of MuSCs into myoblasts. Recent single nucleus RNA-seq (snRNA-seq) and single nucleus ATAC-seq (snATAC-seq) studies on skeletal muscle shed light on the diverse genetic programs of slow, fast, atrophic, and regenerative myofibers[4–7]. Developing and mature myofibers express unique gene programs[8], and several developmental genes can be transcriptionally repressed by electrical activity from motor innervation[9]. However, the transcriptional regulators of the different phases of muscle maturation still need to be defined. This knowledge

[1]Department of Molecular Biology, the Hamon Center for Regenerative Science and Medicine, and Senator Paul D. Wellstone Muscular Dystrophy Specialized Research Center, University of Texas Southwestern Medical Center, 5323 Harry Hines Boulevard, Dallas, TX 75390, USA. [2]Quantitative Biomedical Research Center, Peter O'Donnell Jr. School of Public Health, 5323 Harry Hines Boulevard, University of Texas Southwestern Medical Center, Dallas, TX 75390, USA. [3]Department of Neuroscience, University of Texas Southwestern Medical Center, 5323 Harry Hines Boulevard, Dallas, TX 75390, USA. ✉e-mail: eric.olson@utsouthwestern.edu

deficit presents a significant challenge in understanding the molecular mechanisms that impair myofiber functions in neuromuscular disease.

Here, we generated a multi-omics snRNA-seq and snATAC-seq atlas of skeletal muscle development. Our results reveal the distinct transcriptional heterogeneity of developing and mature myofibers, and highlight the TFs that are active in each stage. In developing myofibers, the TFs, Myogenin, Klf5, and Tead4 cooperate to activate the transcription of developmental muscle genes. In contrast, in mature myofibers we found high expression of Maf, a TF previously unassociated with muscle formation. By ChIP-seq experiments and the analysis of Maf Knockout (KO) mice, we found that Maf is an essential mediator of myofiber maturation. Furthermore, we analyzed mice lacking Cacna1s, which encodes the α1s subunit of voltage-gated L-type $Ca^{2+}$ channel ($Ca_v$1.1) in skeletal muscles. We found that the absence of $Ca_v$1.1 led to impairments in Maf gene expression and myofiber maturation, suggesting that $Ca^{2+}$ influx and excitation-contraction coupling play important roles in activating Maf expression and promoting myofiber maturation. Together, these findings provide a holistic view of the transcriptional underpinnings of muscle development and maturation.

## Results

### A multi-omics snRNA-seq and snATAC-seq atlas of skeletal muscle formation

To provide an integrated view of gene expression and transcriptional regulation of the successive phases of myogenesis, we generated an atlas of snRNA-seq and snATAC-seq from developing mouse skeletal muscles (Fig. 1a). We performed multi-omics experiments to simultaneously profile gene expression and open chromatin landscapes of individual nuclei. Nuclei were isolated from skeletal muscles dissected from the hindlimbs without skin or bones at embryonic day (E) 14.5, fetal (E18.5), neonatal (P5), and adult stages (2 months). After quality control and filtering, we recovered 15,402 nuclei, expressing an average of 2000 genes per nucleus (Supplementary Fig. 1a–e). Nuclei from snRNA-seq data were clustered by Uniform Manifold Approximation and Projection (UMAP) with the Seurat software[10] (Fig. 1b). The identity

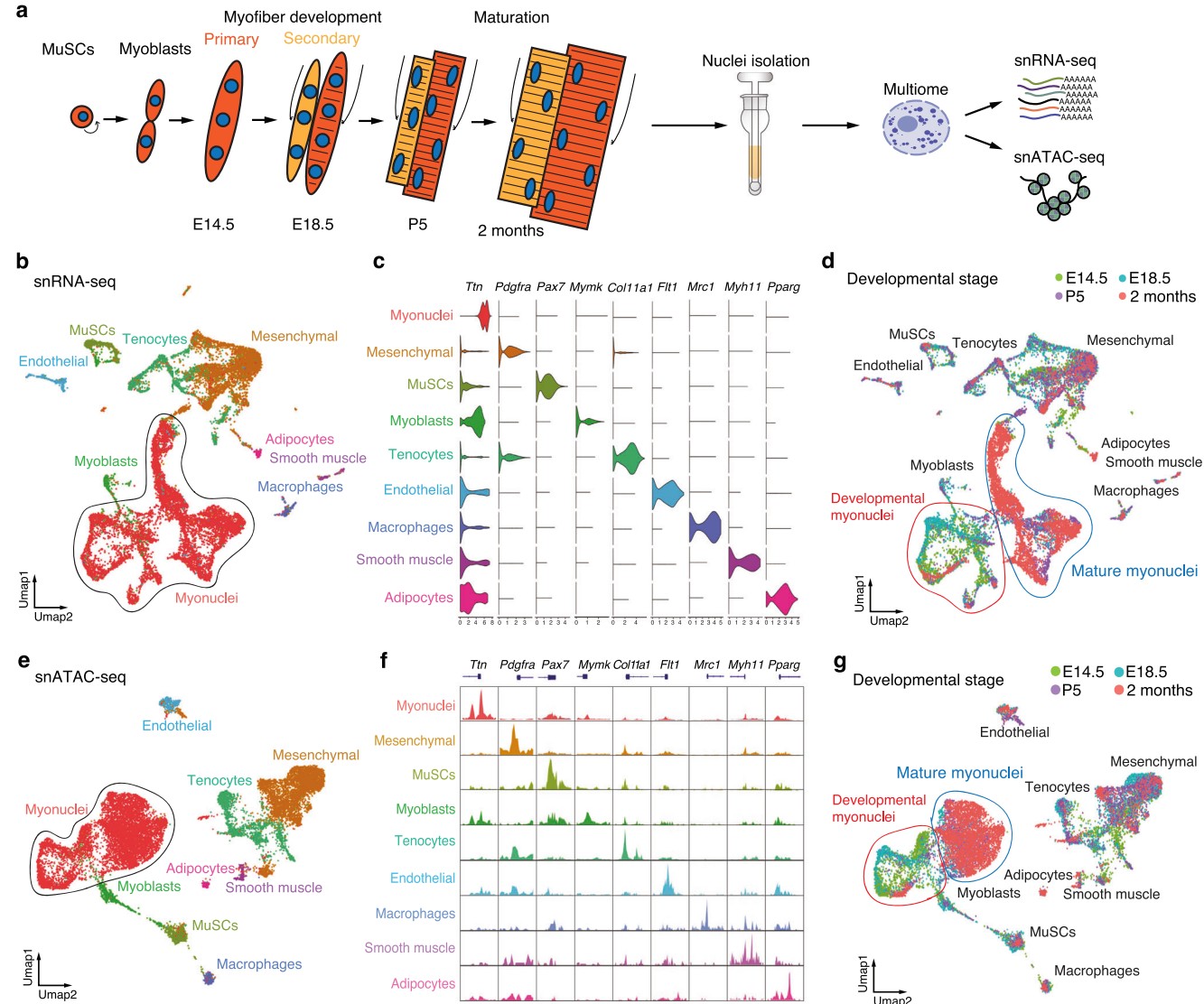

**Fig. 1 | Multi-omics single nucleus RNA-seq and ATAC-seq atlas of mouse skeletal muscle development. a** Schematic of the experimental design for multi-omic snRNA-seq and snATAC-seq of skeletal muscle development. **b** UMAP unsupervised clustering of snRNA-seq data from E14.5, E18.5, P5, and adult skeletal muscle. MuSCs: muscle stem cells. **c** Violin plots showing the expression of marker genes in different cell populations. **d** The same UMAP visualization as in (**b**), showing the time origin of each nucleus. **e** UMAP visualization of snATAC-seq data from E14.5, E18.5, P5, and adult skeletal muscle. **f** Chromatin accessibility of the promoter regions of the marker genes of Fig. 1c, in the different cell populations. **g** The same UMAP visualization of snATAC-seq data as in e, showing the time origin of each nucleus.

of each nuclear cluster was determined by the expression of specific marker genes (Fig. 1c and Supplementary Fig. 1f). We detected nuclei from different cell types, including myofibers (myonuclei), mesenchymal cells, tenocytes, endothelial cells, macrophages, smooth muscle cells, adipocytes, MuSCs and myoblasts.

To document changes in gene expression over time in the different cell populations, we further marked each nucleus according to its developmental stage (Fig. 1d). For most of the clusters, nuclei from different time points clustered together. We detected a population of mesenchymal cells only present at E14.5, E18.5, and P5 but not in adult skeletal muscle. These cells express proliferation marker genes (*Cyclin B1* and *Ki67*) and could correspond to progenitors that can differentiate into fibroblasts, tenocytes, chondrocytes, and osteoblasts (Supplementary Fig. 2a)[11]. Myoblasts were only detected during development and not in adult muscles. Developmental myonuclei (E14.5 and E18.5) clustered separately from mature myonuclei (P5 and adult muscle). Comparison of gene expression in MuSCs, myoblasts, and adult myonuclei identified a total of 4,047 differentially expressed genes (DEGs) (*P*-value < 0.05), revealing a dynamic change in gene expression in myogenic cells over time (Supplementary Fig. 2b).

We then analyzed the snATAC-seq data using the Signac software[12] and clustered nuclei based on 133,110 filtered chromatin accessible peaks (Fig. 1e and Supplementary Fig. 2c). We characterized the identity of each cluster in the snATAC-seq UMAP, based on the marker gene expression from snRNA-seq (Fig. 1c). The assigned classification of the nuclei was confirmed by the specific opening of chromatin in the promoter regions of marker genes of the cell population expressing these genes (Fig. 1f). The same cell populations were observed in the snRNA-seq and snATAC-seq UMAPs. Myoblasts were only detected during development and not in adult muscles. We observed a similar separation of developmental and mature myonuclei in the snRNA-seq (Fig. 1d) and snATAC-seq (Fig. 1g) UMAPs. In summary, these findings provided an atlas of skeletal muscle development and maturation and revealed distinctive patterns of transcription in myogenic cells over time.

### Identification of the gene regulatory programs for myofiber development and maturation

To determine the gene programs regulating myofiber development and maturation, we reclustered myonuclei separately from non-myonuclei cell populations (Fig. 2a, b). We performed Weighted Nearest Neighbor (WNN) analysis to cluster nuclei based on the integration of RNA and ATAC-seq profiles[10]. The shape of the UMAP resembled a rainbow with myonuclei separated into nine different clusters (Clusters 1-9), forming a continuous trajectory of myofiber differentiation (Supplementary Fig. 3a). The DEGs in these clusters revealed a dynamic change for expression of genes encoding sarcomeric (*Acta1* and *Myh*), calcium handling (*Casq2, Camk1d, Ryr3*), and metabolic (*Pfkfb3*) proteins (Fig. 2c). We observed seven different clusters of developmental myonuclei from E14.5 to P5, revealing heterogeneity of developing myofibers. Embryonic E14.5 myonuclei were separated into three clusters (Clusters 1-3). Cluster 1 expressed *Col25a1*, which is necessary for myoblast fusion and formation of neuromuscular junctions (NMJs)[13,14] (Fig. 2d). Cluster 2 expressed *Col22a1*, a marker of myotendinous junction (MTJ) myonuclei[15]. Using single-molecule RNA-FISH (smRNA-FISH), we observed the specific localization of *Col25a1* mRNA in the center of embryonic myofibers and *Col22a1* mRNA in the myotendinous junction (Supplementary Fig. 3b). Cluster 1 likely corresponds to myonuclei localized in the center of embryonic myofibers, whereas Cluster 2 may represent nuclei at the tips of myofibers. Fetal E18.5 myonuclei were separated into two clusters (Clusters 4 and 5), with Cluster 4 expressing *Myomaker*, corresponding to the recently formed secondary myofibers. Cluster 5 corresponds to primary myofibers formed earlier and no longer expressing *Myomaker*. The myosin heavy chain isoforms

*Myh3* and *Myh8* were enriched in embryonic and fetal myonuclei, and were replaced after birth by expression of *Myh2*, *Myh1*, and *Myh4*. Postnatal myonuclei (Clusters 6 and 7) co-expressed developmental genes (*Myh8*) and mature genes (*Myh4*). Cluster 8 corresponded to adult myonuclei expressing *Myh2* and *Myh1*, while Cluster 9 expressed *Myh4*. We validated the switch of gene expression for *Myomaker*, *Myomixer*, *Maf* and *Myh4* in developing and mature myofibers by smRNA-FISH (Supplementary Fig. 3c, d). We also validated the specific expression in developmental fibers of *Fstl4* and *Tmem108*, whose roles are unknown in skeletal muscles. These distinct temporal genetic programs could ultimately modulate myofiber contractile and metabolic properties at each developmental stage.

To identify the TFs that activate these successive gene programs, we integrated snRNA-seq and snATAC-seq data to characterize the TF binding motifs enriched in ATAC peaks (Fig. 2e, f and Supplementary Fig. 3e–g). This analysis revealed waves of expression of TFs that are active during development and are repressed in adulthood or vice versa. We detected TFs with a known role during myogenesis, such as Sox4[16], and Etv5[17] in the embryonic stage, Myogenin[18], Tead4[19], and Klf5[20] in the fetal stage, Prrx1[21] and Nfix[22] in the neonatal stage, and Thrb[23], Esr1[24] and Nr4a1[25] in adult muscle. Interestingly this analysis also revealed the enrichment of TFs with unknown functions during myogenesis, such as Plag1 and Elk3 in embryonic myonuclei, Nfia, Nfib, and Nfic in neonatal myonuclei, and Maf and Tef in adult myonuclei. Together, these results constitute a blueprint of the gene programs and the transcription factors expressed during the successive phases of myofiber development and maturation.

### Myogenin, Klf5, and Tead4 activate the expression of developmental muscle genes synergistically

Myogenin, Tead4, and Klf5 were the most highly expressed and active TFs in fetal myofibers compared to adult myofibers (Fig. 3a). These TFs have been reported to regulate myogenic differentiation individually[18–20], but their possible cooperativity has not been examined. Western Blot analysis revealed the co-expression of Myogenin, Klf5, and Tead4 in developmental hindlimb skeletal muscles and the decreased expression of the 3 TFs in the adult stage (Supplementary Fig. 4a). To test if they cooperate to activate gene expression, we examined peaks co-occupied by these TFs in ChIP-Seq data from myotubes generated in vitro by differentiating the muscle cell line C2C12[19,20,26] (Fig. 3b and Supplementary Fig. 4b). Co-occupied peaks were localized close to developmental muscle genes (Fig. 3c, d). One co-occupied peak was in the promoter of the *Chrng* gene, which encodes a subunit of the acetylcholine receptor, necessary for the function of the NMJ[27]. Another co-bound peak was in the promoter region of *Tnnt2*, encoding an isoform of troponin T required for sarcomere formation[28]. Other peaks were localized next to genes involved in myoblast fusion, like *Col25a1*[13], *Myomaker*[29], and *Myomixer*[30]. All these co-bound peaks contained histone H3K27Ac marks and were specifically active in developmental myonuclei and inactive in adulthood as shown by snATAC-seq (Fig. 3c). To demonstrate that Myogenin, Tead4, and Klf5 positively activate the expression of these developmental muscle genes, we performed knockdown of these TFs in C2C12 cells (differentiated for 3 days) with shRNA lentivirus infection. The expression of *Tnnt2, Col25a1, Myomixer*, and *Myomaker* was significantly reduced by the knockdown of Myogenin, Klf5, or Tead4 (Supplementary Fig. 4c).

To test if Myogenin, Tead4, and Klf5 cooperate synergistically, we cloned the *Myomaker* enhancer that contains binding sites for these three TFs into a luciferase reporter construct upstream of a minimal promoter (Supplementary Fig. 4d). We transfected HEK293T cells with this reporter construct and Myogenin, Tead4, and Klf5 individually or in combination. Expression of each TF alone or as pairs modestly increased luciferase activity (Fig. 3e). In contrast, expression of all three TFs enhanced luciferase activity by ~10-fold. Using a

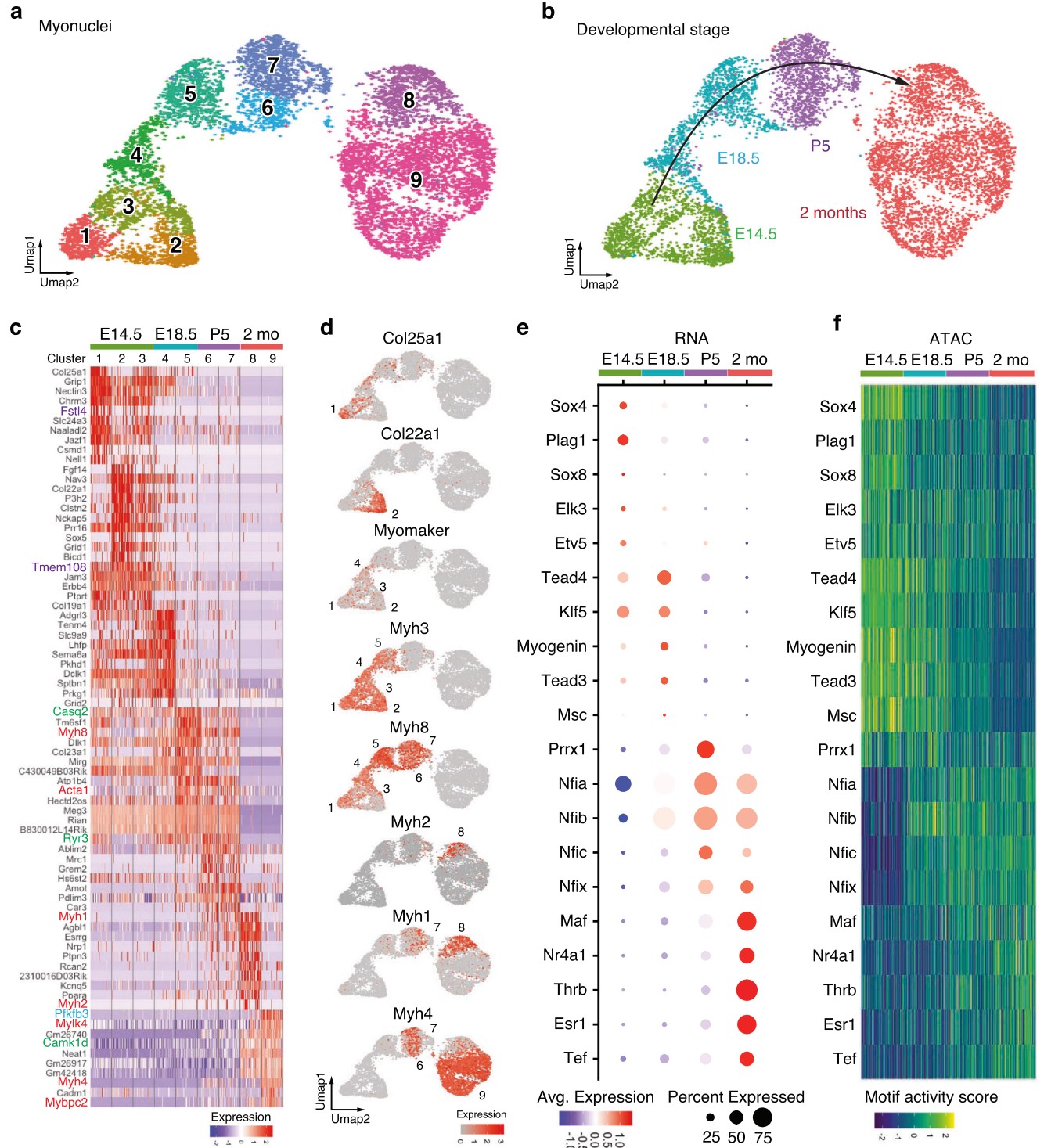

**Fig. 2 | Gene regulatory program of myonuclei maturation. a** UMAP visualization of myonuclei reclustered with the integrative analysis of snRNA-seq and snATAC-seq data by Weighted Nearest Neighbor (WNN). The different colors correspond to the different clusters identified by Seurat. **b** The same UMAP visualization as in (**a**), showing the time origin of each nucleus. **c** Heatmap of the ten most enriched genes in each cluster of myonuclei from Fig. 2a. The values correspond to z-scores of normalized counts. Genes encoding sarcomeric proteins are highlighted in red,

calcium handling proteins in green, and metabolic proteins in blue. **d** UMAPs depicting the expression patterns of *Col25a1*, *Col22a1*, *Myomaker*, and the different isoforms of *Myh* in myonuclei during development and adulthood. **e** Dot-plots of the TFs enriched in mRNA expression in the myonuclei at different development times. **f** Heatmap of TFs enriched in ATAC-seq motif peaks in the myonuclei at different development times.

mutated version of the *Myomaker* enhancer that Myogenin, Tead4, or Klf5 cannot bind, we found that each binding site was essential for the synergistic activation of the enhancer (Fig. 3f and Supplementary Fig. 4e).

To test whether Myogenin, Tead4, and Klf5 interact, we performed co-immunoprecipitation experiments in HEK293T cells expressing tagged versions of these TFs (Fig. 3g). We detected interaction of Myogenin with Tead4 and Klf5 and interaction between Klf5

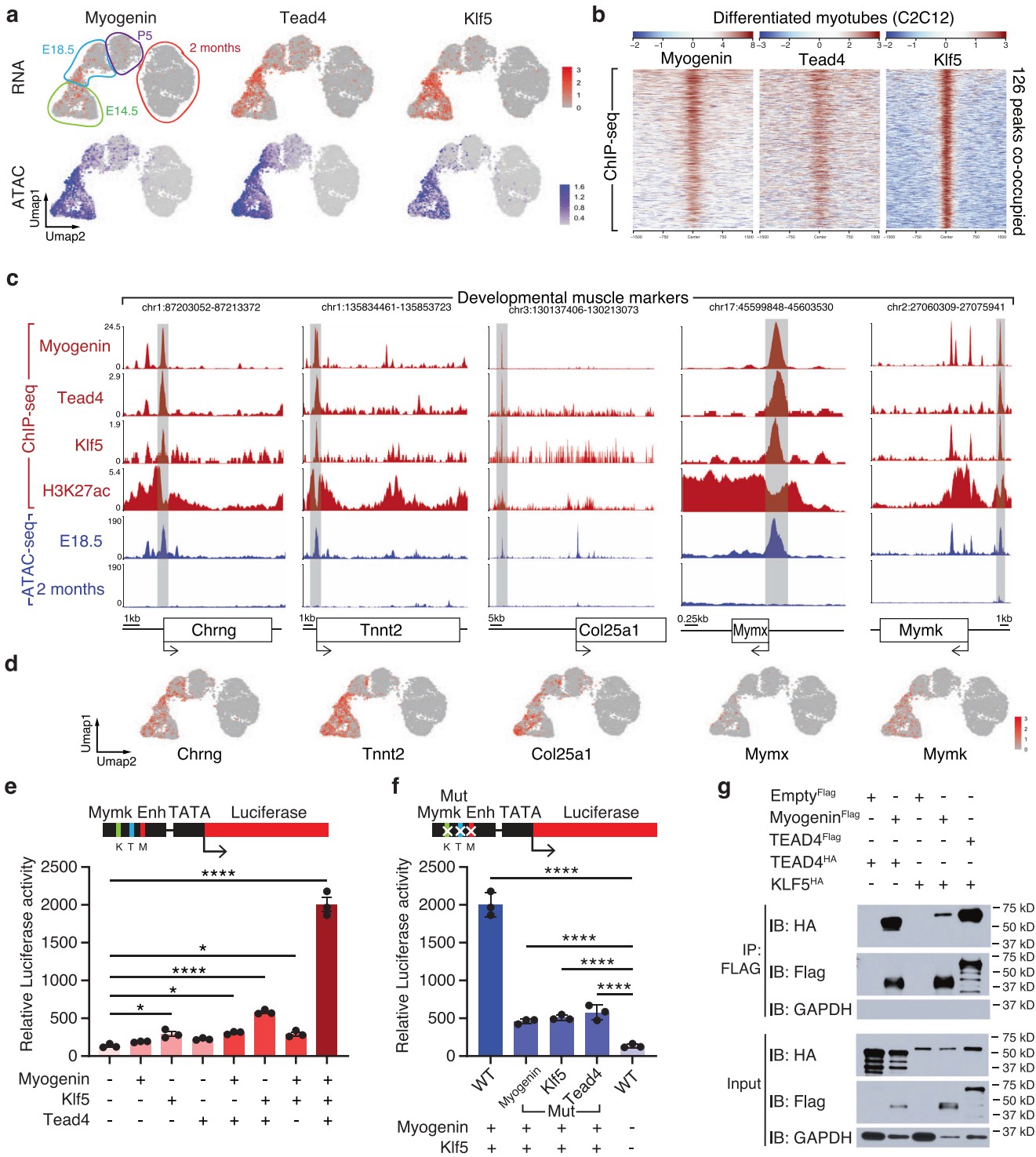

**Fig. 3 | Myogenin, Klf5, and Tead4 activate the expression of developmental muscle genes synergistically. a** UMAPs depicting the mRNA expression in snRNA-seq and activity of Myogenin, Tead4, and Klf5 TFs in snATAC-seq in myonuclei. **b** Heatmaps depicting the Myogenin, Tead4 and Klf5 co-occupied peaks in ChIP-seq data from C2C12 myotubes. **c** Myogenin, Tead4, Klf5 ChIP-Seq, H3K27ac ChIP-Seq, and snATAC-seq tracks depicting co-bound peaks at the *Chrng, Tnnt2, Col25a1, Myomixer (Mymx)*, and *Myomaker (Mymk)* locus. **d** UMAPs depicting the expression of *Chrng, Tnnt2, Col25a1, Myomixer*, and *Myomaker* in myonuclei. **e** Luciferase reporter assays examining Myomaker enhancer activities in HEK293T cells co-transfected with Myogenin, Tead4, and Klf5. Binding sites for Klf5, Tead4, and Myogenin are designated K, T and M, respectively. Luciferase

activity is normalized to LacZ control vector expression. $n = 3$ independent experiments. **f** The same luciferase reporter assays as in Fig. 3e with the Myomaker enhancer mutated in the binding sites of Myogenin, Klf5, and Tead4. $n = 3$ independent experiments. The green rectangle in the Myomaker enhancer corresponds to the mutation of the binding site of Klf5, the red Myogenin, and the blue Tead4. **g** Co-immunoprecipitation and Western blot analysis in HEK293T cells expressing Myogenin-Flag, Tead4-Flag and Klf5-HA, and Tead4-HA (representative results from 2 independent experiments). Numerical data are presented as mean ± s.e.m. *$P < 0.05$, ****$P < 0.0001$. Significance of difference for (**e**, **f**): Two way anova.

and Tead4. We next confirmed the interaction of Myogenin with Tead4 and Tead4 with Klf5 in C2C12 cells, by performing co-immunoprecipitation with endogenous proteins (Supplementary Fig. 4f). These findings indicate that Myogenin, Klf5, and Tead4 form a transcriptional complex that synergistically activates the expression of muscle genes in developing myofibers.

## Maf promotes myofiber maturation

The substantial enrichment of Maf mRNA expression and its transcriptional activity in adult myonuclei prompted us to investigate the role of Maf in myofiber maturation (Fig. 4a). Maf is a basic domain/

leucine zipper (bZIP) transcription factor that plays important roles in the development of the eye[31,32], sensory neurons[33], bone cells[34], and immune cells[35]. In adult mice, Maf is highly expressed in fast skeletal muscle compared to other organs (Fig. 4b). Western blot and snRNA-seq data from adult skeletal muscle revealed higher expresssion of Maf mRNA and protein in fast skeletal muscle fibers compared to slow (Fig. 4c and Supplementary Fig. 5a). Previous observations showed the specific enrichment of Maf in myogenic cells differentiated in vitro and a recent article characterized Maf as a major regulator of fast muscle fibers[36,37]. Based on the expression pattern and transcriptional activity of Maf, we surmised that it plays a role in skeletal muscle maturation.

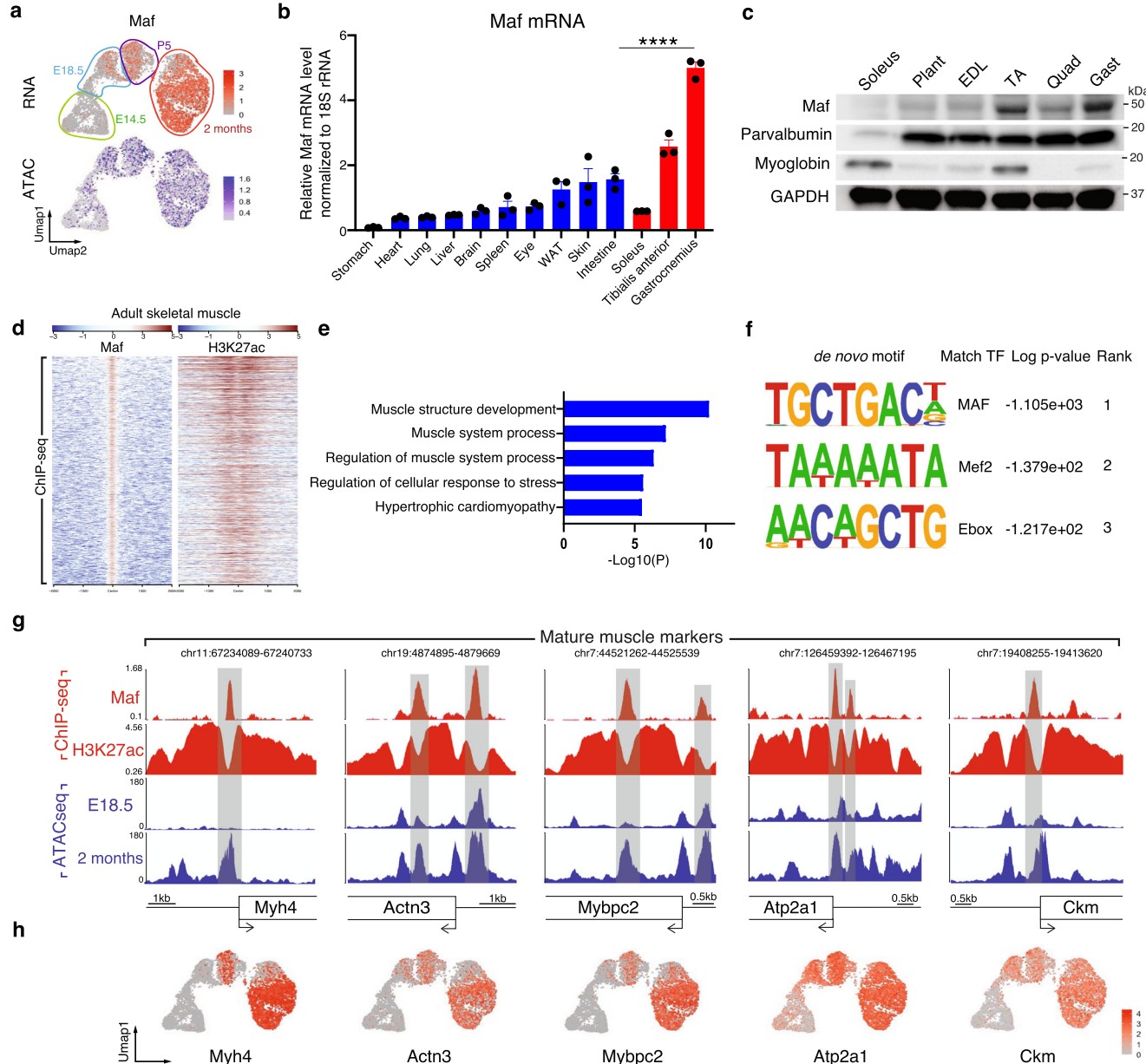

**Fig. 4 | Maf activates fast mature muscle gene expression. a** UMAPs depicting Maf mRNA expression and its motif activity in myonuclei. **b** Expression profiles by RT-qPCR of Maf in diverse adult mice organs ($n = 3$ mice, ****$P < 0.0001$). WAT: White adipose tissue. **c** Western blot showing the accumulation of Maf protein in fast skeletal muscle in adult WT mice. GAPDH is a loading control (representative results from 2 independent experiments). Plant: Plantaris, EDL: Extensor digitorum longus, TA: Tibialis anterior, Quad: Quadriceps, Gast: Gastrocnemius. **d** Heatmaps depicting the Maf peaks and the corresponding H3K27ac peaks in ChIP-seq-data from adult gastrocnemius and quadriceps skeletal muscle. **e** Top GO terms enriched in genes regulated by Maf. **f** De novo motifs identified at Maf bound sites in adult skeletal muscle. *P*-values were calculated by hypergeometric enrichment with HOMER without adjustments. **g** Maf ChIP-Seq, H3K27ac ChIP-Seq and ATAC-Seq tracks depicting peaks at the *Myh4, actn3, Mybpc2, Atp2a1*, and *Ckm* locus. **h** UMAPs depicting the expression of *Myh4, Actn3, Mybpc2, Atp2a1* and *Ckm* in myonuclei by snRNA-seq. Numerical data are presented as mean ± s.e.m. ****$P < 0.0001$. Significance of difference for b: One way ANOVA with adjusted *P*-value.

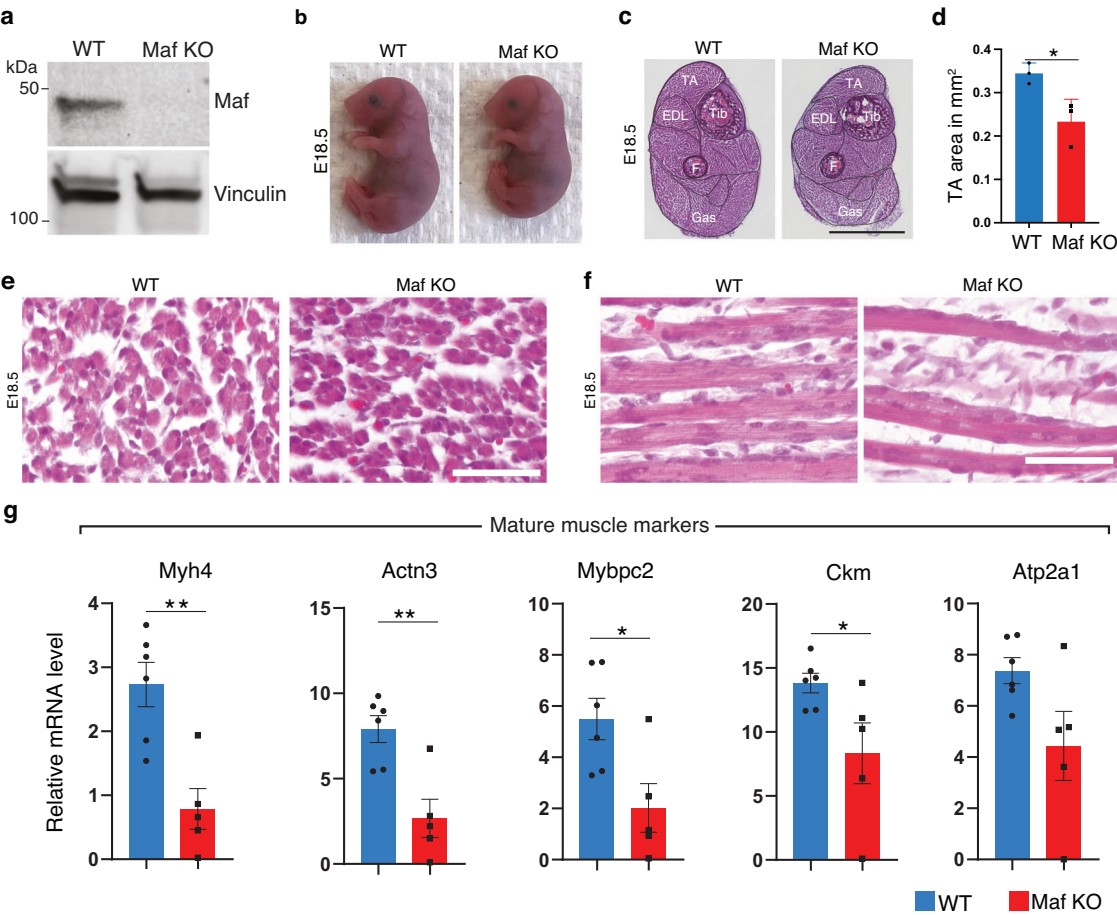

**Fig. 5 | Maf is required for myofiber maturation. a** Western blot showing the absence of Maf protein expression in hindlimb skeletal muscle in Maf KO mice. Vinculin is a loading control (representative results from 2 independent experiments). **b** External appearance of WT and Maf Knockout (KO) mice at E18.5. **c** H&E staining of hind limb transverse sections of WT and Maf KO E18.5 mice. Scale bar: 1 mm. TA: Tibialis anterior. EDL: Extensor digitorum longus. Tib: Tibia. F: Fibula. Gas: Gastrocnemius. **d** Quantification of TA surface area in Maf KO compared to WT (*n* = 3 mice per group, \**P* = 0.0406). **e** H&E staining of transverse WT and Maf KO E18.5 myofibers from TA (representative results from 3 independent experiments). Scale bar: 62.5 μm. **f** H&E staining of longitudinal WT and Maf KO E18.5 myofibers from vastus lateralis (representative results from 3 independent experiments). Scale bar: 62.5 μm. **g** Expression level of the mature genes *Myh4, Actn3, Mybpc2, Ckm* and *Atp2a1* in WT and Maf KO hindlimb skeletal muscle as detected by RT-qPCR. The absence of Maf impairs the expression of the mature program (*n* = 6 mice for WT, and *n* = 5 mice for Maf KO). Numerical data are presented as mean ± s.e.m. \**P* < 0.05, \*\**P* < 0.01. Significance of difference for (**d**, **g**): Two-tailed, unpaired t-test.

To determine the direct transcriptional targets of Maf in myofibers and compare these targets with the active promoter and enhancer landscape, we performed ChIP-seq using antibodies against Maf and H3K27Ac in adult gastrocnemius and quadriceps skeletal muscles (Fig. 4d). We detected 5456 Maf binding peaks (FDR = 0.001). Among them, 3793 peaks were associated with positive H3K27ac marks, indicating that Maf is mainly a transcriptional activator in skeletal muscle. Gene Ontology analysis of these peaks revealed that the associated genes were involved in skeletal muscle function (Fig. 4e). Motif analysis using Homer[38] revealed that in addition to the canonical Maf binding motifs, the Maf peaks possessed binding motifs such as the Mef2 motif and E-boxes, suggesting the cooperativity of Maf with Mef2 and myogenic bHLH factors (Fig. 4f). Among the loci with the strongest binding of Maf, we detected promoters of mature muscle genes including *Myh4, Actn3, Mybpc2, Atp2a1*, and *Ckm* (Fig. 4g). Interestingly, these genes are only expressed during myofiber maturation and show the same expression pattern as Maf (Fig. 4a, h). Consistent with this finding, developmental muscle genes do not contain Maf binding peaks (Supplementary Fig. 5b).

Previous studies showed that Maf-deficient mice exhibited defects in the differentiation of lens fiber cells and die before weaning[31,32]. However, possible skeletal muscle phenotypes have not

been explored in these KO mice. To address this, we generated Maf knockout (KO) mice by introducing a premature stop codon and deleting a large region of the coding sequence using CRISPR/Cas9 technology (Supplementary Fig. 5c, d). Western blot analysis confirmed the complete loss of Maf protein expression in the hindlimb skeletal muscles of the Maf KO mice (Fig. 5a). As previously described in refs. [31,32], Maf KO mice did not survive after birth. At E18.5, Maf KO embryos were smaller and displayed severe kyphosis, as seen in other mouse mutants lacking developmental muscle genes[18,39] (Fig. 5b). Histological analysis of hind limbs revealed a reduction in muscle size across different muscle groups in Maf KO mice compared to WT (Fig. 5c). Quantification of the tibialis anterior (TA) area at three different section levels revealed a diminution of 32% of TA area in the absence of Maf (Fig. 5d). Transverse and longitudinal sections showed that Maf KO myofibers were multinucleated but appeared smaller compared to WT (Fig. 5e, f).

To determine the cause of muscle hypoplasia in Maf KO mice, we quantified mRNA expression of developmental and mature muscle genes. We found that expression of the mature genes, such as *Myh4, Actn3, Mybpc2, Atp2a1*, and *Ckm*, was dramatically impaired in the absence of Maf (Fig. 5g). Maf directly bound the promoter regions of these mature muscle genes to activate their expression (Fig. 4g). In

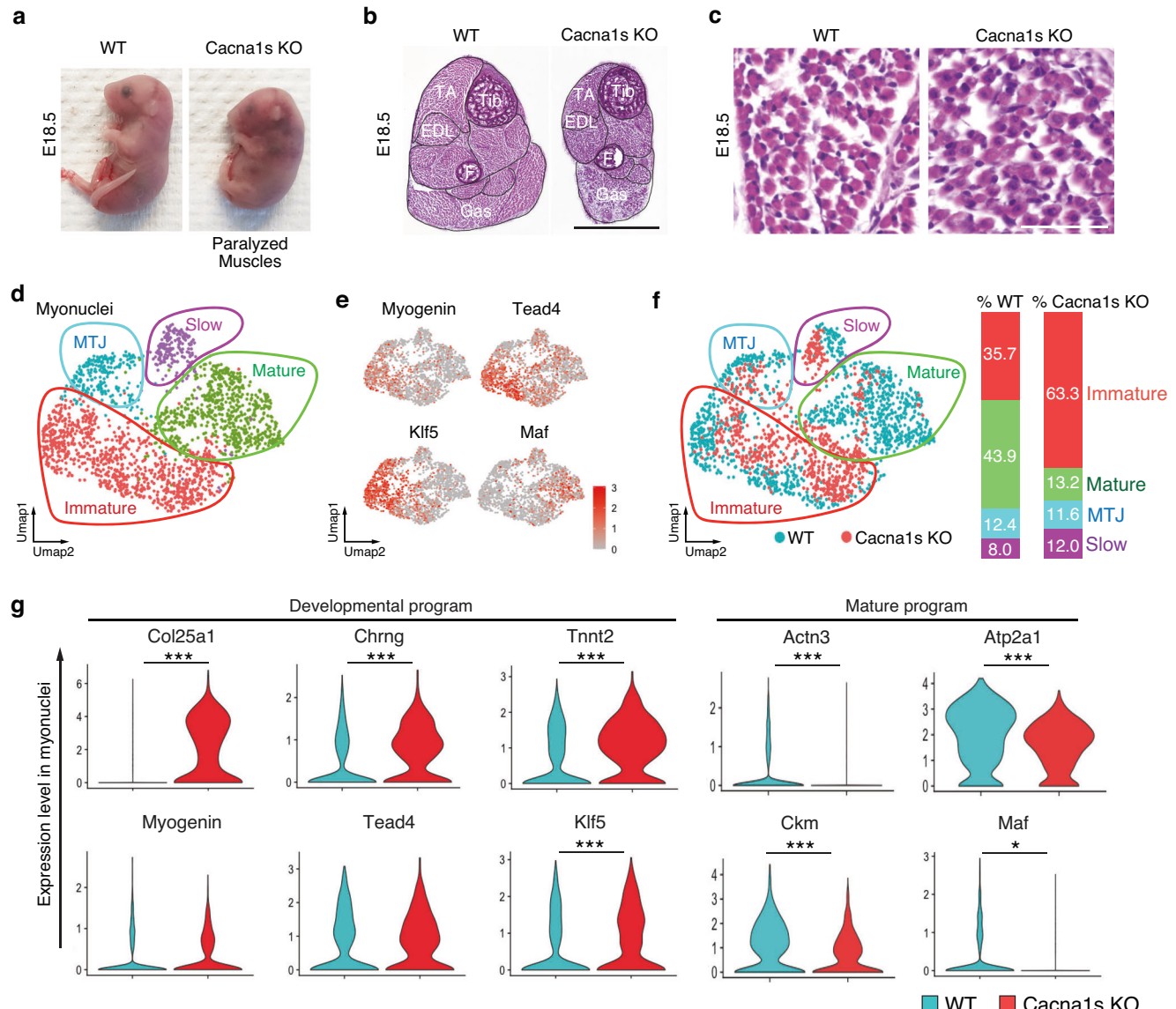

**Fig. 6 | Suppression of Maf gene expression and myofiber maturation in Cacna1s KO mice. a** External appearance of WT and Cacna1s Knockout (KO) mice at E18.5. **b** H&E staining of hind limb transverse sections of WT and Cacna1s KO mice at E18.5 (representative results from 3 independent experiments). Scale bar: 1 mm. TA: Tibialis anterior. EDL: Extensor digitorum longus. Tib: Tibia. F: Fibula. Gas: Gastrocnemius. **c** H&E staining of transverse WT and Cacna1s KO E18.5 myofibers from TA (representative results from 3 independent experiments). Scale bar: 62.5 μm. **d** UMAP visualization of myonuclei reclustering. The different colors correspond to the different clusters identified by Seurat. **e** The same UMAPs visualized in (**d**), showing the expression of several markers used to identify the different myonuclei subtypes. **f** The same UMAP visualized in (**d**), showing the WT nuclei in blue and Cacna1s KO nuclei in red. Percentages of myonuclei subtypes in WT and Cacna1s KO muscles are indicated on the right. **g** Expression of *Col25a1, Chrng, Tnnt2, Actn3, Atp2a1, Ckm, Myogenin, Tead4, Klf5,* and *Maf* in WT and Cacna1s KO myonuclei.*P < 0.05, ***P < 0.001. Significance of difference for (**g**): Two-tailed, non-parametric Wilcoxon.

contrast, expression of the muscle developmental gene program (*Chrng, Tnnt2, Mymk, Col25a1, Myh7, Myh3, and Myh8*) and TFs (*Myogenin, Klf5,* and *Tead4*) was similar in WT and Maf KO muscles at RNA and protein levels (Supplementary Fig. 5e, f). These findings revealed that Maf promotes the maturation of myofibers by activating the mature gene program.

### Maf gene expression and myofiber maturation are suppressed in skeletal muscles lacking Ca$_v$1.1

Myofiber contraction is triggered by a cascade of events known as excitation-contraction (EC) coupling initiated by depolarization of the muscle membrane, Ca$^{2+}$ influx through Ca$_v$1.1, the release of Ca$^{2+}$ from the sarcoplasmic reticulum into the cytoplasm via the opening of the ryanodine receptor and activation of the sarcomeres[40]

(Supplementary Fig. 6a). Intracellular Ca$^{2+}$ also modulates the activity of several signaling pathways, and transcriptional activity of adult myofibers[41]. However, it is not well understood how EC coupling affects the transcriptional activity of developing myofibers and the maturation of myofibers. We therefore analyzed skeletal muscles from mutant mice deficient in EC coupling due to the absence of *Cacna1s*, which encodes the α1s subunit of Ca$_v$1.1 (Fig. 6a and Supplementary Fig. 6b). Previous studies have shown that action potential-induced Ca$^{2+}$ transients and EC coupling are absent in muscular dysgenic (*mdg*) mice deficient in *Cacna1s*[42–46]. Similar to *mdg* mice, Cacna1s KO mice were paralyzed and died at birth due to inability to breathe. At E18.5, KO embryos were smaller and hunchbacked. Hematoxylin & eosin staining of hind limbs of Cacna1s KO mice revealed a substantial reduction in muscle area across different muscle groups compared to

WT muscles (Fig. 6b, c). We also found a decrease in the number of myofibers, which contributed to muscle hypolasia (Fig. 6b, c and Supplementary Fig 6c). The remaining myofibers in the KO mice expressed developmental muscle genes, including *Myh7, Myh3*, and *Myh8* (Supplementary Fig. 6c).

We then performed snRNA-seq and snATAC-seq multi-omics analysis on E18.5 Cacna1s KO and WT hindlimb skeletal muscles (Fig. 6d and Supplementary Fig. 6d,e). We observed significant changes in cell composition in Cacna1s KO muscle, such as MuSCs, myoblasts, adipocytes, and tenocytes, compared to WT muscle (Supplementary Fig. 6f). These changes are due to the loss of Cacna1s in myonuclei, since the expression of *Cacna1s* is restricted to myonuclei and no other cell types (Supplementary Fig. 6g).

To focus our analysis on myofibers, we clustered myonuclei separately and identified four distinct myonuclei clusters: MTJ (*Col22a1*), slow fibers (*Myh7*), immature fibers expressing *Myogenin*, *Tead4*, and *Klf5*, and mature fibers expressing *Maf* (Fig. 6d, e and Supplementary Fig. 7a). There was an ~2-fold increase in the percentage of immature myonuclei and a 3-fold decrease in the percentage of mature myonuclei in the Cacna1s KO muscle compared to WT muscle (Fig. 6f). The immaturity of KO fibers was further confirmed by the increased expression of the developmental genes *Col25a1*, *Chrng*, and *Tnnt2* and the decreased expression of mature genes *Actn3*, *Atp2a1*, and *Ckm* in Cacna1s KO myonuclei compared to WT (Fig. 6g and Supplementary Fig. 7b, c). Expression of the developmental TFs *Myogenin*, *Klf5*, and *Tead4*, was also slightly increased in Cacna1s KO myonuclei. Conversely, expression of Maf, which controls the mature gene program, was significantly decreased. Interestingly, around half of the genes downregulated in Cacna1s KO are associated with binding peaks of Maf in ChIPseq data (Supplementary Fig. 7d), suggesting that Maf directly regulates their gene expression. Western blot experiments confirmed the increased expression of the developmental TFs Myogenin, Klf5, and Tead4 and the decreased expression of Maf in paralyzed skeletal muscle (Supplementary Fig. 7e).

Cacna1s KO myonuclei showed changes in gene expression analogous to denervated adult muscle fibers, such as down-regulation of *Hdac9* and up-regulation of *Hdac4, MuSK, Chrna1, Chrnb1, Chrnd, Chrng, Foxo1, Foxo3*, and *Trim63* (Supplementary Fig. 7f), suggesting a common mechanism of gene regulation by muscle contraction during development and adulthood. Altogether, these results revealed that the skeletal muscle L-type Ca²⁺ channel is required for myofiber maturation by repressing the transcriptional activity of Myogenin, Klf5, and Tead4 and activating the expression of Maf (Fig. 7).

## Discussion

Despite numerous studies of muscle gene expression, gene regulatory programs that activate the successive phases of myogenesis in vivo have not been fully described. In this study, we generated a snRNA-seq and snATAC-seq multi-omics atlas of myogenesis from both developmental and adult mouse skeletal muscle. Our data revealed several waves of gene regulatory programs and TFs regulating the differentiation of MuSCs into myoblasts and the development of embryonic, fetal, neonatal, and adult myofibers. Furthermore, analyzing paralyzed muscles in Cacna1s KO mice shed light on how electrical activity from motor nerves represses developmental and activates mature gene programs in myofibers.

Our results show that Myogenin, Tead4, and Klf5 are the most active TFs in developing myofibers compared to mature myofibers. The requirement of these TFs during myogenesis has been demonstrated by the impairment of muscle differentiation when silencing either Myogenin, Tead4, or Klf5 in vitro and in vivo[18–20]. Regulation of gene expression frequently requires collaboration of multiple TFs. By analyzing ChIP-seq data, we found that Myogenin, Tead4, and Klf5 cooperate to activate the expression of genes required for myoblast fusion, as well as the formation of sarcomeres and the NMJ in myoblasts and developing myofibers. The concerted action of these TFs allows synergistic activation of gene expression by forming a transcriptional complex. This complex could determine the robust spatial and temporal gene expression pattern in cells that express the three TFs simultaneously. This complex could also integrate several signaling pathways to finely control myogenic differentiation. This transcriptional complex could also cooperate with other myogenic TFs expressed at all phases of myofiber development and maturation, such as Mef2 TFs. After birth, the expression and activity of this transcriptional complex are repressed, and analysis of Cacna1s KO muscles revealed that this repression depends on Ca²⁺ influx from the L-type Ca²⁺ channel. Previous studies showed that electrical activity represses *Myogenin* gene expression[9] and Myogenin transcriptional activity via CaMKII phosphorylation[47]. A similar repression mechanism by electrical activity and Ca²⁺ signaling could reduce the expression and activities of Tead4 and Klf5 during myofiber maturation.

Maf is highly expressed and active in mature myofibers, compared to developmental stages, but its role in myofiber formation has not been previously explored. Maf is not expressed in muscle progenitor cells (MuSCs and myoblasts), and it begins to be expressed and active in a subset of fetal myofibers (Cluster 5). In these cells, Maf promotes myofiber maturation by directly activating the expression of the mature gene program, as seen by the decreased expression of mature

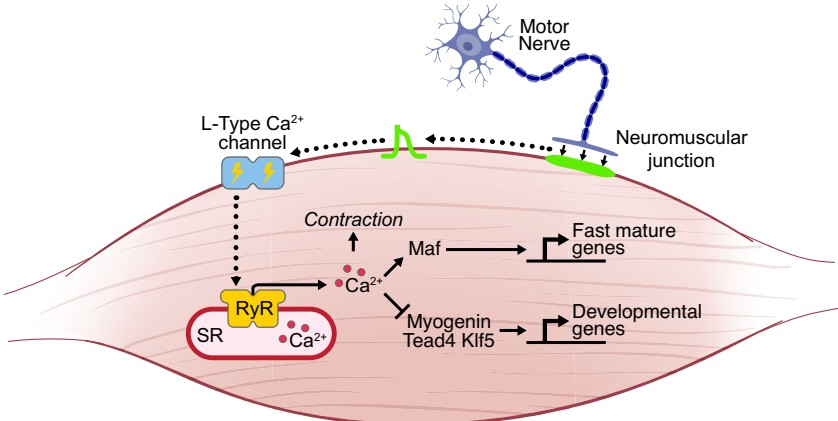

**Fig. 7 | Genetic program regulating the transition from developmental to mature myofibers.** In newly formed muscle fibers, transcription factors Myogenin, Tead4 and Klf5 are activated to express developmental genes. During muscle fiber maturation, L-type Ca²⁺ channel activity and excitation-contraction coupling repress Myogenin, Tead4, Klf5 and the developmental gene program, and activate Maf. By activating the transcription of fast-mature genes, Maf induces final maturation of muscle fibers. Abbreviations: Sarcoplasmic reticulum (SR).

genes (*Myh4, Actn3, Mybpc2, Atp2a1,* and *Ckm)* and the muscle hypoplasia in E18.5 Maf KO muscles.

In adult myofibers, *Maf* mRNA and protein are more expressed in fast Myh4+ myofibers compared to other myofibers, and in snATAC-seq data, Maf binding motif is also more active in fast Myh4+ myofibers than other myofibers[4]. The ChIP-seq analysis of Maf in adult skeletal muscle revealed specific binding of Maf in the Myh4 promoter and not in the Myh1, Myh2, or Myh7 promoters (Supplementary Fig. 5A). A recent article by Sadaki et al. reported that large Maf transcription factors (Mafa, Mafb, and Maf) are important regulators of fast Myh4 myofibers[37].

Maf controls the expression of the fast mature gene program in response to $Ca^{2+}$ signaling and muscle contraction. In other tissues, Maf activates the transcription of genes essential for the formation of the eye[31], sensory neurons[33], bone cells[34], and immune cells[35]. In these tissues, several signaling pathways regulate the activity and expression of Maf. For example, in T lymphocytes, Maf expression is induced by Il6 stimulation via calcineurin activation and Stat3 transcriptional activity[48]. A similar $Ca^{2+}$ dependent signaling pathway could activate the expression of Maf and myofiber maturation.

Developmental genes are frequently reexpressed in diseased muscle. This reexpression of developmental genes can serve as a compensatory response to repair damaged myofibers or promote the reinnervation of myofibers. However, several reports indicate that reexpression of the developmental gene program can be detrimental to adult myofiber integrity[49–51]. By revealing TFs that regulate myofiber development, our study provides a deeper understanding of the pathways that are deregulated in disease.

Myogenin expression is upregulated in denervated muscle and participates in the expression of atrophic genes[50]. Tead4 and Klf5 are also upregulated in atrophic muscle[52,53]. In cooperation with Myogenin, Tead4 and Klf5 could synergistically activate the expression of an atrophic gene regulatory program in diseased myofibers. Conversely, the expression and activity of Maf are diminished in denervated myofibers[54]. This diminution could decrease the expression of mature fast muscle genes during atrophy. In Schwann cells, histone deacetylase 5 (Hdac5) represses Maf expression[55]. Interestingly, Hdac5 KO muscles are protected from denervation-induced atrophy[50], and it would be interesting to determine if Maf expression is unchanged by denervation in these muscles. The developmental TFs Myogenin, Tead4, Klf5, and the mature TF Maf could control opposing genetic programs regulating muscle properties during development and disease. These findings provide a new perspective for the development of therapies ameliorating muscle functions in neuromuscular diseases.

## Methods

### Animals
Animal work described in this manuscript was approved and conducted under the oversight of the University of Texas (UT) Southwestern Institutional Animal Care and Use Committee (IACUC). Mice were housed in a pathogen-free barrier facility with a 12-h light/dark cycle, with a temperature of 18–24 °C and humidity of 35–60%, and maintained on standard chow (2916; Teklad Global). For WT mice, we used C57BL/6 N from Jackson laboratory. To generate Maf[+/-] mice, we designed 4 sgRNAs targeting the first exon of the Maf gene. Pronuclear injection of sgRNAs and Cas9 mRNA was performed as previously described in ref. 56. Cacna1s[+/-] mice (Cacna1s[tm1.1(KOMP)Vlcg], MGI:5494476) were generated by the KOMP-Regeneron project, through an insertion of the Velocigene cassette ZEN-Ub1, which led to a deletion of the coding region of the Cacna1s gene[57]. We obtained Cacna1s[+/-] mice from Mutant Mouse Resource & Research Centers (MMRRC) supported by the NIH at UC Davis (MMRRC:046868-UCD). Pregnant females were staged, taking the appearance of the vaginal plug as E0.5. For Multiomics snRNA-seq and snATAC-seq experiments, embryos and fetuses were harvested at 14.5 and 18.5 days post-

fertilization. P5 pups were harvested five days after birth. For adult conditions, we used 8 week-old male and female C57black6n mice.

### Nuclei isolation for single nucleus RNA-seq and single nucleus ATAC-seq
Nuclei were extracted using a modified version of a previously described protocol in ref. 58. For all stages, hindlimb skeletal muscles (without skin and bone) were dissected, snap-frozen, and stored in liquid nitrogen. For E14.5 embryos, 56 hindlimbs were pooled together. For E18.5 embryos, 14 hindlimbs were pooled together. For P5 mice, 8 hindlimbs were pooled together. For adult mice, 4 hindlimbs were pooled together. For E18.5 Cacna1s KO mice, 26 hindlimbs were pooled together. Skeletal muscles were thawed for 5 min on ice. Muscles were minced with scissors in 1 ml cold lysis buffer (10 mM Tris-HCl pH7.4, 10 mM NaCl, 3 mM MgCl2, 0.1% NonidetTM P40, 1 mM DTT, 1 U/µl RNase inhibitor in Nuclease-Free Water). After 2 min, 4 ml of cold lysis buffer was added, and muscles were lysed for 5 min at 4 °C. 10 ml of cold wash buffer (PBS, BSA 2%, and 0.2 U/µl RNase inhibitor from Roche) was added, and the lysate was dounced with 20 strokes of a loose pestle. The homogenate was filtered with 70 µm and 40 µm cell strainers. Nuclei were pelleted by centrifugation for 5 min at 500 × g at 4 °C. Nuclei were resuspended in 0.5 ml of wash buffer and 2 µl of PCM1 antibody (Sigma-Aldrich, HPA023374) to stain the myonuclei. After 15 min, nuclei were pelleted by centrifugation for 5 min at 500 × g at 4 °C. Nuclei were resuspended in 0.5 ml of wash buffer and 2 µl of an Alexa 647 conjugated PCNA antibody (Biolegend, 682203) to stain all the nuclei and 1 µl of Alexa 555 Goat anti-rabbit secondary antibody against PCM1. After 15 min, nuclei were pelleted by centrifugation for 5 min at 500 × g at 4 °C. Then the nuclei were resuspended in 300 µl of wash buffer and filtered with 30µm cell strainers.

### FACS sorting of nuclei
Nuclei were FACS sorted to exclude debris with a BD FACSAria III and the BD FACSDIVA and FlowJo v10.8.1 software. PCM1 positive, and PCNA positive (myonuclei) were sorted in one 15 ml tube and PCM1 negative and PCNA positive nuclei were sorted in another 15 ml tube. More than 100,000 nuclei were collected for each condition. PCM1 positive and negative nuclei were mixed in equal volumes and pelleted by centrifugation for 10 min at 500 × g at 4 °C to analyze myonuclei and other cell types. Nuclei were then resuspended in 200 µl of ATAC lysis buffer (10 mM Tris-HCl pH7.4, 10 mM NaCl, 3 mM MgCl2, 0.01% Tween-20, 0.01% NonidetTM P40, 0.001% Digitonin, 1% BSA, 1 mM DTT, 1U/ul RNase inhibitor in Nuclease-Free Water). After 2 min, 1.3 ml of ATAC wash buffer (10 mM Tris-HCl pH7.4, 10 mM NaCl, 3 mM MgCl2, 0.1% Tween-20, 1 mM DTT, 1% BSA, 1U/µl RNase inhibitor in Nuclease-Free Water) was added. Nuclei were pelleted by centrifugation for 10 min at 500 × g at 4 °C and resuspended in 20 µl of diluted nuclei buffer from 10X genomics.

### Multi-omic snRNA-seq and snATAC-seq
Nuclei concentration was quantified with an automatic hemocytometer, and was adjusted to 2000 nuclei/µl with wash buffer. We loaded around 5000 nuclei per condition into the 10x Chromium Chip. We then used the Next GEM single-cell Multiome ATAC+ Gene expression kit according to the manufacturer's protocol. GEM-incubation was performed in a thermal cycler: 37 °C for 45 min, 25 °C for 30 min. Post GEM-incubation Cleanup using DynaBeads MyOne Silane Beads was followed by Pre-Amplification PCR (72 °C for 5 min, 98 °C for 3 min, cycled 7 × 98 °C for 20 s, 63 °C for 30 s, 72 °C for 1 min). After cleanup with SPRIselect Reagent Kit, the ATAC library was generated (98 °C for 45 s, cycled 8 × 98 °C for 20 s, 67 °C for 30 s, 72 °C for 20 s) and purified with SPRIselect Reagent Kit. The fragment size estimation of the resulting ATAC libraries was assessed with Agilent TapeStation D1000 HS (Agilent) and quantified using the QubitTM

double stranded DNA (dsDNA) High-Sensitivity HS assay (Thermo-Fisher Scientific). SnATAC-seq libraries were then sequenced by pair with a HighOutput flowcell using an Illumina Nextseq 500 with the following mode: Read 1 N − 63 cycles, i7 index − 8 cycles, i5 index − 24 cycles *(8 dark cycles and 16 cycles on i5), Read 2 N − 63 cycles. SnRNA-seq libraries were constructed by performing the following steps: cDNA Amplification, SPRIselect cleanup, Fragmentation, End repair and A tailing, SPRIselect cleanup, and Adaptor ligation, SPRIselect cleanup, Samplex index PCR and SPRIselect cleanup size selection. The fragment size estimation of the resulting libraries was assessed with Agilent TapeStation D1000 (Agilent) and quantified using the QubitTM double stranded DNA (dsDNA) High-Sensitivity HS assay (Thermo-Fisher Scientific). SnRNA-seq libraries were then sequenced by pair with a HighOutput flowcel using an Illumina Nextseq 500 with the following mode: pair-end 28 bp (read1) + 10 bp (index1) + 10 bp (index2) + 110 bp (read2).

## Multiome data analysis

For snRNA-seq, more than 30,000 reads per nucleus were sequenced. For snATAC-seq, more than 20,000 reads per nucleus were sequenced. To analyze jointly the gene expression and ATAC measurements, we used Cell Ranger Arc software 2.0.0 from 10x Genomics. Raw base call files (BCL) from the Nextseq 500 were demultiplexed with the cellranger-arc mkfastq pipeline into FASTQ files. The same command was used to demultiplex both ATAC and GEX flow cells. The FASTQ files were then processed with the Cell Ranger ARC count pipeline to perform alignment, filtering, barcode counting, peak calling and counting of both ATAC and GEX molecules with parameters set to default. We used a reference genome built against mouse mm10 (Sequence: GRCm38 Ensembl 93), and included intronic reads in our analysis. Reads associated with retained barcodes were quantified and used to build a transcript count table. Visualizations, clustering, and differential expression tests were performed in R version 4.1.0 using Seurat 4.0.4 and Signac 1.4.0. Quality control was performed by keeping cells with > 400 and < 25,000 nCount RNA, cells with <10% mitochondrial genes and nCount ATAC < 7e4. We obtained 3799 nuclei from E14.5, 3784 nuclei from E18.5, 3235 nuclei from P5, 4584 nuclei from adult and 4238 nuclei from Cacna1s KO E18.5 muscles. We detected expression of approximately two thousand genes per nucleus in each condition (Supplementary Fig. 1). Pathway analysis was performed using Metascape[59]. TFs motif activity score was analyzed by running chromVAR.

## ChIP-seq

Skeletal muscle nuclei were isolated as previously described in ref. [60] and ChIP-seq was carried out as previously described in ref. [61], with modifications. Nuclei were purified from 2-month-old WT gastrocnemius and quadriceps and crosslinked with 4% formaldehyde (Sigma) in PBS for 10 min, then quenched with 0.125 M glycine for 10 min at room temperature. Crosslinked samples were then washed with cold PBS. Samples were collected by a brief spin and treated with 10 mM Tris-HCl (pH 8.0), 10 mM NaCl and 0.2% NP-40 for 30 min to collect nuclei. After nuclear extraction, chromatin was sheared on a Bioruptor Pico (Diagenode) for 10 cycles (30 s on/ 30 s off for each cycle) at 4 °C in sonication buffer (0.1% SDS, 1% Triton X-100, 10 mM Tris-HCl, 1 mM EDTA, 0.1% sodium deoxycholate, 0.25% sarkosyl, 1 mM DTT, 1x Complete Protease Inhibitor Cocktail (Roche), and 200 μM PMSF, pH 8.0). After sonication, 1% of the sonicated chromatin from each sample was taken out as 'Input' samples. The remaining sonicated chromatin was evenly split for Maf or H3K27ac ChIP. NaCl was then added to a final concentration of 300 mM for histone ChIP and 150 mM for transcription factor ChIP. 1 μg/mL H3K27ac antibody (Diagenode, C15410196) or 10 μg/mL Maf antibody (Thermofischer, A300-613A) was added to each sample and incubated at 4 °C overnight with gentle rotation. The next day, 30 μL/mL of pre-washed

Dynabeads Protein G (Invitrogen, 10004D) was added to each sample for a two-hour incubation. After that, the beads were washed twice with 1 mL RIPA 0 buffer (0.1% SDS, 1% Triton X-100, 10 mM Tris-HCl, 1 mM EDTA, 0.1% sodium deoxycholate, pH 8.0), twice with 1 mL RIPA 0.3 buffer (0.1% SDS, 1% Triton X-100, 10 mM Tris-HCl, 1 mM EDTA, 0.1% sodium deoxycholate, 300 mM NaCl, pH 8.0), twice with 1 mL LiCl wash buffer (250 mM LiCl, 0.5% IGEPAL CA-630, 0.5% sodium deoxycholate, 1 mM EDTA, 10 mM Tris-HCl, pH 8.0), and finally twice with 1 mL TE buffer (10 mM Tris-HCl, 1 mM EDTA, pH 8.0). For each ChIP sample or input sample, 100 uL of SDS elution buffer (1% SDS, 10 mM EDTA, 50 mM Tris-HCl, pH 8.0) was added and incubation was done at 65 °C overnight on a ThermoMixer (Eppendorf) at 1000 rpm. The next day, the supernatant was collected and further treated with 0.5 μg RNaseA (Sigma, 11119915001) for 30 min at 37 °C, followed by 20 μg Proteinase K (NEB, P8107S) treatment at 37 °C for 2 h. DNA was recovered using MinElute PCR Purification Kit (QIAGEN, 28004) according to the manufacturer's protocol. The quality of the raw NGS data was checked with multiqc v1.7 and filtered with Samtools v0.1.19 and Sambamba v0.066. Reads adaptors were trimmed with trimgalorev0.6.4, then aligned into the genome reference using Bowtie2 v2.3.4.3 and Star v2.7.3a. Bam files were generated with Bedtools v2.29.2. Motif analysis proceeded with HOMER v4.4 and heatmap visualized with deepTolls v2.0.

## Luciferase assay

The DNA sequence of the Myomaker enhancer was synthetized by IDT and PCR amplified using GXL polymerase. The enhancer was cloned into a pGL4.Luc plasmid using KpnI and HindIII restriction enzymes. PE cells were transfected with the Enhancer reporter construct, CMV-Lacz plasmid and plasmids encoding the TFs in triplicate. Cells were lysed in Passive Lysis Buffer (Dual-Luciferase Reporter Assay System, Promega) and frozen for 15 min at −80 °C. After thawing, cells were kept on ice and lysates were resuspended 7 times by pipetting. Lysates were transferred into a 96 well plate (white bottom) with of reconstituted renilla luciferase (from Promega). Renilla luciferase activity was measured by plate reader (BMG Labtech, CLARIOstar). After measurement, lysates were stained against Beta-galactosidase and quantified with the same plate reader. Beta-galactosidase measurements were normalized for all conditions to the negative control. Then luciferase signal was normalized to the normalized Beta-galactosidase value for that condition.

## Co-immunoprecipitation

Co-immunoprecipitation (CoIP) was performed as previously described in ref. [62], with modifications. Three days post-infection, cells were scraped in ice-cold PBS and lysed in 800 μL of lysis buffer (50 mM Tris-HCl at pH 7.5, 150 mM NaCl, 1% NP-40, 0.1% sodium deoxycholate, 2× protease inhibitor tablet [Roche]) for 30 min on ice and cleared by centrifugation at 14,000 g for 10 min at 4 °C. 50 μL of lysates were saved for 'Input'. For each pull-down, 2 μg of Flag M2 antibody (Sigma) was bound to 20 μL of washed Dynabeads Protein G (Invitrogen). Lysates were then added to antibody-coupled beads. Reactions were incubated on a rotating platform for 3 h at 4 °C, and then beads were washed five times with 1 mL of wash buffer (50 mM Tris-HCl at pH 7.5, 150 mM NaCl, 0.05% NP-40, 2× protease inhibitor tablet [Roche]). Immunoprecipitated proteins were eluted with 40 μL of elution buffer (36 μL of lysis buffer supplemented with 4 μL of 3× Flag peptide at 10 mg/mL [Sigma]) with shaking at 1000 RPM for 30 min at room temperature. Immunoblotting for Input and CoIP samples was performed as described earlier with the following antibody: Tead4 (Abcam, ab58310, dilution 1/1000), Klf5 (Proteintech, 21017-1-AP, dilution 1/1000), Myogenin (Santa Cruz Biotechnology, sc-12732, dilution 1/1000), Goat anti-Rabbit IgG (H + L)-HRP conjugate (Bio-Rad, 1706515, dilution 1/5000) and Goat anti-Mouse IgG (H + L)-HRP conjugate (Bio-Rad, 1706516, dilution 1/1000).

## Quantitative real-time PCR analysis

qPCR reactions were assembled using PowerTrack SYBR Green Master Mix (Applied Biosystems). Assays were performed using Applied Biosystems QuantStudio 5 Real-Time PCR System (Applied Biosystems). Expression values were normalized to 18 S mRNA. Sequences of oligonucleotides are provided in Supplementary Information.

## Histology

Hindlimbs were dissected out and submerged in 4% paraformaldehyde in PBS overnight at 4 °C, followed by PBS wash. Hindlimbs were then incubated overnight at 15% sucrose/PBS and 30% sucrose/PBS at 4 °C. Tissu were then embedded in O.C.T (Fisher) and sectioned at 10 μm with a cryostat (Leica CM1860). Immunostaining and RNAscope were performed as previously described in ref. 63. The following primary antibodies were used: Myh3 (Developmental Studies Hybridoma Bank, F1.652, concentration 1/200), Myh8 (Developmental Studies Hybridoma Bank N3.36, concentration 1/200), Myh7 (Developmental Studies Hybridoma Bank BA-F8, dilution 1/40). The following secondary antibody were used Alexa555 goat α-mouse IgG1 (ThermoFischer scientific, A-21127, dilution 1/1000), Alexa647 goat α-mouse IgM (ThermoFischer scientific, A-21238, dilution 1/1000).

## Immunoblotting

Western blot was performed as described for Co-immunoprecipitation with the following antibody: Maf (Abcam, ab77071, dilution 1/1000), Vinculin (Sigma-Aldrich, V9131, dilution 1/1000), Tead4 (Abcam, ab58310, dilution 1/1000), Klf5 (Proteintech, 21017-1-AP, dilution 1/1000), Myogenin (Santa Cruz Biotechnology, sc-12732, dilution 1/1000), Parvalbumin (Millipore Sigma, MAB1572, clone name: PARV-19, dilution 1/1000), Myoglobin (Santa Cruz Biotechnology, sc-7425, dilution 1/1000).

## Cell lines

C2C12 and HEK293T cell lines were obtained from American Type Culture Collection (CRL-1772, CRL-3216). All cell lines were authenticated by short tandem repeat analysis (STR) profiling and positively matched. Cell lines were controlled for mycoplasm contamination using the Universal Mycoplasma Detection Kit (ATCC) and were tested negative.

## Statistics & reproducibility

All values are given as mean ± SEM. Differences between two groups were assessed using unpaired two-tailed Student's t-tests, and differences between multiple groups were assessed using one way Anova tests. $P < 0.05$ was regarded as significant. Statistical analysis was performed in Graphpad. No statistical method was used to predetermine sample size. No data were excluded from the analyses. The experiments were not randomized. The investigators were not blinded to allocation during experiments and outcome assessment.

## Data availability

SnRNA-seq, snATAC-seq and ChIP-seq data are available in the NCBI Gene Expression Omnibus (GEO) database under accession number GSE211545. Antibodies used in the study are provided in Supplementary Table 1 and Oligonucleotides used for RT-qPCR experiments in the Supplementary Table 2. Source data are provided with this paper.

## Code availability

The script used for the analysis of single nuclei RNAseq and single nuclei ATACseq data is available here: https://github.com/matthieudossantos/Multiome-snRNAseq-and-snATACseq-Muscle-development-atlas-script

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

## Acknowledgements

We thank Drs. Jian Xu and Yoon Jung Kim for Illumina sequencing, the Moody Foundation Flow Cytometry from Children's Medical Research Institute for nuclei sorting, and UT Southwestern Histology Core for H&E staining. We are grateful to Jose Cabrera for graphic assistance. We thank Drs. Francesco Chemello, Xurde M. Caravia and Zhaoning Wang

(University of California San Diego) for helpful discussion. This work was supported by funds from NIH (AR071980, HL130253, HL157281, and NS055028), the Senator Paul D. Wellstone Muscular Dystrophy Specialized Research Center (P50 HD087351), and the Robert A. Welch Foundation (grant 1-0025).

## Author contributions

M.D.S., R.B-D., N.L. and E.N.O. designed the experiments and overall study and wrote the manuscript. M.D.S, J.R.M generated the Maf KO mouse models. M.D.S., A.M.S., S.B. and Y.Z. performed experiments. W.L. designed the experiments on Cacna1s KO mice. M.D.S., K.C. and L.X. did the bioinformatic analysis. All authors discussed the results and participated in the article preparation and editing.

## Competing interests

The authors declare no competing interests.
