## [Peer Review File · Nature Communications]

Opposing gene regulatory programs governing myofiber development and maturation revealed at single nucleus resolutionREVIEWER COMMENTS

Reviewer #1 (Remarks to the Author):

In this study, Dos Santos and colleagues are performing a transcriptional mapping of myofibers' maturation, from the embryo to the adult skeletal muscle, at the single nucleus resolution (RNAseq and ATACseq). Their analysis identifies previously unknown stage-specific transcriptional profiles and suggests, as the authors describe, that distinct genetic programs are operating to modulate myofiber properties at each developmental stage. Focusing on transcription factors, they identify a trimeric complex consisting of Myogenin, Tead4 and Klf5 that operates in foetal myofibers, and Maf in the adult myonuclei. To investigate further the role of Maf, the authors generated a knock-out mouse and performed ChIPseq with MAF and H3K27ac antibodies in adult hindlimb muscles. Lastly, analysis is performed on the paralysed muscles of the Ca²⁺ channel Cacna1s KO mice. Cacna1s is not derived from the analysis described above, but the authors justify it as a model to investigate the role of neural stimulation and myofiber maturation. Indeed, it is shown that Maf expression is strongly impaired accompanied by a reduction of other transcripts linked to myofibre maturation.

The study is well performed and it yields many novel and interesting data. Moreover, it establishes a developmental trajectory of myonuclei maturation and identifies Maf as an important transcription factor in this process. However, despite the conceptual significance, this study is weak in functional validations. Moreover, although the use of mutant mice is appreciated, the genetic tools used are not optimal since they are lacking spatiotemporal regulation. Below are some specific comments that could strengthen this study.

-The snRNAseq/ATACseq experiments for the embryonic/foetal stages seem to have been performed on whole hindlimbs and the adult experiments on dissected muscles. EC, adipocytes and mesenchymal cells and macrophages are, therefore, not muscle specific. Even if this is not the main scope of the study, it should be more clearly noted in the text, especially in the context of the integrated Umaps.

-It is recommended to validate some of the most interesting candidate genes on muscle sections by FISH experiments. RNAscope or equivalent should be performed for a few stage-specific transcripts to validate their developmental dynamics and assess the spatial distribution (e.g. tip or central domain, as suggested for Clusters 1, 2).

-On page 5 the authors conclude that "These distinct temporal genetic programs ultimately modulate myofiber contractile and metabolic properties at each developmental stage". This is an over-statement since the transcriptome of the myonuclei could be linked to other properties than the ones referred to here, or even the outcome of 2ary responses that only specific experiments could validate.

- Linked to the analysis on the TFs that *activate* the successive gene programs, on page 6 the authors conclude: "Together, these results constitute a blueprint of the gene programs and the transcription factors regulating the successive phases of myofiber development and maturation". Once again, this statement is not backed up by functional experiments so it cannot be conclusive. The TFs were identified based on DE analysis of mRNA levels and the enriched binding motifs in ATAC peaks. Although this is a powerful method, it is not providing information on the actual transcriptional activity of the TFs. The identification of previously known TFs, like Sox4 and Nfix, is a partial validation of the approach but still, the new factors remain untested. This comment may appear obvious to the authors, and it is true that essentially all sc/snRNAseq studies do include a section of candidate genes, simply based on differential expression. Here, the authors go further by performing a more detailed analysis of 3 TFs, Myog/Tead4/Klf5, identified as enriched in

foetal myofibers. They analyse ChIPseq data in myotubes generated by C2C12 cells (which are adult, not foetal) and find promising regulatory elements that seem to be simultaneously co-occupied by the 3 TFs (note that since this done on bulk, it is not proven that all TFs bind the same element in a single cell). Since single-cell ChIPseq is not possible and ChIP on ChIP experiments are too demanding, some functional experiments should be performed to validate the cooperative function of the 3 TFs on the candidate targets. To this end, the authors perform luciferase assays on one cloned enhancer close to Myomaker and nicely demonstrate cooperative action. Classical gain and lof experiments in myogenic cells, however, should also be performed and the effect on target genes by RT-qPCR should be measured (Myomaker and ideally others). Without these functional experiments, the conclusion that “Myogenin, Tead4, and Klf5 positively activate the expression of these genes” is not supported (note that the association of regulatory elements to genes is based on proximity for practical analytical reasons; without a transcriptional response to TF manipulations, the association between a TF and a gene is inadequate).

-The muscle phenotype in the Maf KO mice is very interesting. Moreover, the specific deregulation of mature genes and not of genes linked to the developmental program is excellent. One aspect that is probably worth discussing is that Maf was identified as a TF of adult myofibers yet the KO mice have a clear muscle phenotype at E18.5. Also, it should be explained why you made a new KO mouse since one is already available. It is unfortunate that the new Maf allele is not Cre-inducible, which would allow spatiotemporal regulation (and study specifically the role of adult Maf).

Less major and editorial comments:

-Please, give more information on the samples that were analysed for the snRNAseq/ATACseq experiments. It is unclear in the Methods if the P5 sequencing was done on whole hindlimbs or dissected muscles.

-“ The UMAP resembled a rainbow with myonuclei separated into nine different clusters”. What is the scientific meaning of a UMAP resembling a rainbow? Is this metaphor meaningful in some way?

- Page 9 “The decreased expression of these mature genes reflects the lack of expression of Maf in Cacna1s KO myofibers”. Do you mean “The decreased expression of these mature genes IS REFLECTED BY the lack of expression of Maf in Cacna1s KO myofibers”?

-You may want to reconsider the title as, in my view, Maf is not the main finding of this study.

Reviewer #2 (Remarks to the Author):

In the current manuscript, Matthieu et al., performed snRNA-seq and snATAC-seq of skeleton muscle cells undergoing maturation from perinatal to postnatal stage. During perinatal development, they demonstrate that Myogenin, Klf5, and Tead4 cooperatively activate cardiac genes and facilitate the maturation of myocytes. At the postnatal stage, the authors identified Maf as a novel transcription factor to activate the maturation program. They further discovered the L-type Calcium channel regulates the myocyte maturation by repressing the activity of Myogenin, Klf5, and Tead4, and activating Maf. Overall, the dataset generated in current manuscript is a valuable resource for the understanding of skeleton muscle development. However, there are several major concerns needed to be addressed before the publication of the manuscript.

Major:

1. The bioinformatic analysis performed on the multi-omics dataset is quite minimal. More in-depth analyses, even using standard and established algorithms, could greatly benefit the manuscript. For example, the authors could reconstruct pseudotime trajectory or calculate RNA velocity to have a continuous alignment of cells undergoing maturation. Such analysis would allow them to explore the genes with different expression patterns along the maturation process without bias.
2. In Figure 2, it is unclear how the enriched motifs were identified from the multi-omic analysis. The methods are missing in the method section. If the authors used chromVar, please include the motif activity score alongside the expression heatmap (Figure 2e) to justify their selection of the motifs to follow up.
3. In Figure 3, the overlapped peaks in only 0.2% of the total peaks, this is too low. The authors need to perform statistical analysis to test whether such overlap is significant or is merely odds (insignificant thus lacks biological meaning). Also, within the 126 peaks, are these peaks mainly associated with the myocyte maturation? GO analysis would be helpful.
4. As the authors used an in vitro differentiation of C2C12 to validate the function of Myogenin, Klf5, and Tead4, it is critical to know if the binding of these TFs correlate with the open chromatin accessibility during muscle maturation from their scATAC-seq. Additionally it is recommended to provide the information if there is a change in DNA accessibility along the muscle maturation on these TFs' binding peaks.
5. The last part of the manuscript seems to be disjoint from the previous section of the manuscript. A better transition or justification is needed to switch from the multi-omics data analyses to Cacna1s KO experiment.
6. Along the same line, the conclusion made from the bioinformatics analysis (before Cacna1s KO part) lacks support both in silicon and experimentally. To establish the link between the Cacna1s and Maf, the authors may consider analyzing how many Mad target genes are dysregulated in Cacna1s KO muscle cells. Similarly, the slight increase in the expression of Myogenin, klf5, and Tead4 after Cacna1s KO is not sufficient to support they act downstream of Cacna1s. To support such claim, a rescue experiment is needed.

Minor:

1. For Figure 2c legend, it should be top ten instead of top five marker genes in each cluster. Please double check.
2. In Extended Figure 3c, UMAP for Klf5 is missing.
3. Please add the genomic coordinate to Figure 3C, 4F.
4. In Figure 5g, these genes are all the direct targets of Maf. Does the author also observe changes of cardiac maturation genes which are not directly bound by Maf?

Reviewer #3 (Remarks to the Author):

During development, defined gene regulatory networks govern sequential steps of myogenesis, from the specification of myogenic progenitors, to differentiating myoblasts that exit the cell cycle and begin the process of fusion to generate the multinucleate myofibre. However, the transcriptional regulators of muscle maturation remain undefined. While it is known that innervation and electrical stimulation of the myofibre promotes maturation, less is known how innervation of the myofibre modifies transcriptional activity.

In this manuscript, Dos Santos et al. present data delineating the formation and maturation of skeletal muscle based on gene expression programs revealed by combined single-nucleus RNA-sequencing and single-nucleus ATAC-sequencing. They analyzed and compared embryonic (E14.5), fetal (E18.5), neonatal (P5), and adult (2 months) hindlimb skeletal muscle. The authors propose a model of developmentally regulated muscle genes by the synergistic activity of the transcription factors Myogenin, Klf5, and Tead4. They further show that the mature myofiber gene expression program is controlled by the transcription

factor Maf. Maf knockout mice die perinatally and display signs of disturbed skeletal muscle development. Moreover, the authors use a *Cacna1s* knockout mouse model, which is characterized by paralyzed and atrophic muscle, an immature myofiber transcriptome, to demonstrate that Maf expression is reduced upon defective upstream motor innervation.

There are many strengths in this manuscript. Overall, the manuscript presents high quality data obtained from state of the art nucleotide sequencing. The single nuclear RNA-sequencing and single nuclear ATAC-sequencing datasets will be a significant contribution to the myogenesis community. Moreover, the subsequent demonstration of how these snRNA-seq and snATAC-seq datasets can be analyzed to reveal novel regulators of the myogenic program (Myogenin-Klf5-Tead4 \diamond Maf) is excellent. The use of mouse genetic tools provides, in my opinion, the cleanest demonstration of the importance of Maf for myofibre maturation. Overall, the authors provide an interesting model by which the Ca²⁺ transport activity of *Cacna1s* promotes Maf-dependent myofibre maturation, while repressing a Myogenin-Klf5-Tead4 regulatory network that is predominant in muscle development.

The authors interesting model can be better supported by addressing weaknesses in the manuscript as follows:

Major points (revisions needed)

1. The authors rely heavily on their snRNA-seq and snATAC-seq datasets to demonstrate coexpression of Myogenin with Klf5 and Tead4. The work would be better supported by protein analyses, for example immunofluorescence for these TFs in fetal vs adult myofibres.

2. The authors provide some evidence that Myogenin-Tead4-Klf5 transcription factors cooperate to regulate gene expression. In addition to these TFs co-occupying peaks, the authors provide some evidence of interaction in 293 cells. Synergistic transcriptional activity is demonstrated with a myomaker enhancer-promoter luciferase assay in 293 cells. While the presented data is strong and should be included, there is a lack of data supporting the interactions in myogenic cells. These interactions are potentially revealed in fetal myoblasts or at the correct myogenic stages using proximity ligation assay (or similar strategy) to reveal interactions in situ and/or immunoprecipitation of endogenous complexes from myoblasts. As the myofibre matures, one might expect loss of these interactions, which can be reported with the same tools. If feasible, investigating these interactions in the Maf KO and *Cacna1s* KO would also support Figure 5, Figure 6 and the authors presented model (see also point 4 below).

3. The use of mouse genetic tools in the manuscript is a strength of the manuscript, but the solitary use of H&E staining for analysis, nor any quantification of the analyses, does not lead to high quality data. Moreover, the H&E data presented do not convincingly demonstrate a muscle phenotype. The analyses would be improved for example using immunofluorescence with antibodies against Myogenin, embryonic myosin heavy chain, perhaps Myh3 and fast Myh8 (see for example Figure 4B, slow vs fast muscle Maf expression levels). If antibodies are available, immunofluorescence or immunoblotting analyses for genes shown by the authors to be regulated by MAF (eg/ Myh4, Actn3) would support the authors conclusions that MAF regulates myofibre maturation.

4. Figure 6, the model would be better supported if the authors show *Cacna1s* KO embryos have reduced MAF protein expression, potentially shown by immunoblotting as done in Figure 5A or by immunofluorescence analyses if the antibody is suitable. Likewise, increased myogenin KLF5, TEAD4 protein expression could be shown using immunoanalyses (see also the end of point 2 above). I understand the authors present high quality nucleotide sequencing data for these genes, but in my opinion these datasets are starting points for deeper investigation.

5. There is no mechanistic insight into how Ca²⁺ influx from CACNA1S regulates Maf and/or Myogenin-Klf5-Tead4 expression. If known Ca²⁺ responsive regulators such as NFAT or CaMK pathways are not involved, I understand that this point may require deeper investigation outside the context of experimental revisions.

6. Figure 4b needs to be presented with error bars, sample size and statistics.

Minor points (do not necessarily require experimental revisions)

1. Figure 4B, the authors reveal increased Maf mRNA expression in fast vs slow muscle groups, but do not comment. If these changes are significant, is MAF more important for transcription of fast myosin heavy chains vs slow?

2. The authors provide evidence that MAF is required for maturation of the myofibre, but do not comment on whether MAF is needed for maintenance of the myofibre. It would be interesting to know if Maf/MAF expression is decreased upon denervation of adult muscle, or perhaps on single EDL myofibres that have been isolated from the motor neuron, fresh isolated or cultured for short periods of time.

3. Figure 1. Was the identified MuSC cluster unique to adult muscle? I initially had some concerns with respect to labeling myogenic progenitors present at developmental stages as MuSCs. In Figure 1A, should the MuSC be labeled at 2 months with the mature myofibres? Some clarification would help.

REVIEWER COMMENTS

Reviewer #1 (Remarks to the Author):

In this study, Dos Santos and colleagues are performing a transcriptional mapping of myofibers' maturation, from the embryo to the adult skeletal muscle, at the single nucleus resolution (RNAseq and ATACseq). Their analysis identifies previously unknown stage-specific transcriptional profiles and suggests, as the authors describe, that distinct genetic programs are operating to modulate myofiber properties at each developmental stage. Focusing on transcription factors, they identify a trimeric complex consisting of Myogenin, Tead4 and Klf5 that operates in foetal myofibers, and Maf in the adult myonuclei. To investigate further the role of Maf, the authors generated a knock-out mouse and performed ChIPseq with MAF and H3K27ac antibodies in adult hindlimb muscles. Lastly, analysis is performed on the paralysed muscles of the Ca²⁺ channel Cacna1s KO mice. Cacna1s is not derived from the analysis described above, but the authors justify it as a model to investigate the role of neural stimulation and myofiber maturation. Indeed, it is shown that Maf expression is strongly impaired accompanied by a reduction of other transcripts linked to myofibre maturation.

The study is well performed, and it yields many novel and interesting data. Moreover, it establishes a developmental trajectory of myonuclei maturation and identifies Maf as an important transcription factor in this process. However, despite the conceptual significance, this study is weak in functional validations. Moreover, although the use of mutant mice is appreciated, the genetic tools used are not optimal since they are lacking spatiotemporal regulation. Below are some specific comments that could strengthen this study.

Response: Thank you for your enthusiasm and insightful comments. We have addressed your comments and concerns as detailed below and revised our manuscript accordingly.

-The snRNAseq/ATACseq experiments for the embryonic/foetal stages seem to have been performed on whole hindlimbs and the adult experiments on dissected muscles.

EC, adipocytes and mesenchymal cells and macrophages are, therefore, not muscle specific. Even if this is not the main scope of the study, it should be more clearly noted in the text, especially in the context of the integrated Umaps.

Response: The Multi-omic snRNA-seq and snATAC-seq experiments were performed on dissected skeletal muscles from hindlimbs without skin, bones, and adipose tissues at all stages (embryonic, fetal, neonatal, and adult). Therefore, endothelial cells, adipocytes, mesenchymal cells, and macrophages are originating from skeletal muscles. Similar cell composition of skeletal muscle has been reported in other studies (Dos Santos et al 2020., Kim et al 2020., Chemello et al 2020, Petraný et al., 2020). We have clarified this in the Results and Materials and Methods of our revised manuscript (pages 4 and 14 of the revised manuscript).

-It is recommended to validate some of the most interesting candidate genes on muscle sections by FISH experiments. RNAscope or equivalent should be performed for a few stage-specific transcripts to validate their developmental dynamics and assess the spatial distribution (e.g. tip or central domain, as suggested for Clusters 1, 2).

Response: As suggested, we performed smRNA-FISH experiments on isolated hindlimb muscle fibers at different stages to visualize *Col22a1*, *Col25a1*, *Myomaker*, *Myomixer*, *Maf* and *Myh4* mRNA. We observed a specific localization of *Col22a1* in the myotendinous junction and *Col25a1* in the center of embryonic myofibers, confirming the different spatial distribution of clusters 1 and 2 from Figure 2a. Similar to the snRNA-seq data, *Myomaker*, and *Myomixer* were expressed in developmental but not mature myofibers. On the contrary, *Maf* and *Myh4* were specifically expressed in mature but not developmental myofibers. We included these data in revised Supplemental Figures S3b-c.

-On page 5 the authors conclude that “These distinct temporal genetic programs ultimately modulate myofiber contractile and metabolic properties at each developmental stage”. This is an over-statement since the transcriptome of the myonuclei could be linked

to other properties than the ones referred to here, or even the outcome of 2ary responses that only specific experiments could validate.

Response: We agree with this comment and revised the manuscript accordingly to correct this overstatement (page 5 of the revised manuscript).

- Linked to the analysis on the TFs that *activate* the successive gene programs, on page 6 the authors conclude: “Together, these results constitute a blueprint of the gene programs and the transcription factors regulating the successive phases of myofiber development and maturation”. Once again, this statement is not backed up by functional experiments so it cannot be conclusive.

Response: Following the reviewer’s comment, we revised the manuscript accordingly and corrected this overstatement (page 6 of the revised manuscript).

The TFs were identified based on DE analysis of mRNA levels and the enriched binding motifs in ATAC peaks. Although this is a powerful method, it is not providing information on the actual transcriptional activity of the TFs. The identification of previously known TFs, like Sox4 and Nfix, is a partial validation of the approach but still, the new factors remain untested. This comment may appear obvious to the authors, and it is true that essentially all sc/snRNAseq studies do include a section of candidate genes, simply based on differential expression. Here, the authors go further by performing a more detailed analysis of 3 TFs, Myog/Tead4/Klf5, identified as enriched in foetal myofibers.

They analyse ChIPseq data in myotubes generated by C2C12 cells (which are adult, not foetal) and find promising regulatory elements that seem to be simultaneously co-occupied by the 3 TFs (note that since this done on bulk, it is not proven that all TFs bind the same element in a single cell). Since single-cell ChIPseq is not possible and ChIP on ChIP experiments are too demanding, some functional experiments should be performed to validate the cooperative function of the 3 TFs on the candidate targets. To this end, the

authors perform luciferase assays on one cloned enhancer close to *Myomaker* and nicely demonstrate cooperative action. Classical gain and lof experiments in myogenic cells, however, should also be performed and the effect on target genes by RT-qPCR should be measured (*Myomaker* and ideally others). Without these functional experiments, the conclusion that “Myogenin, Tead4, and Klf5 positively activate the expression of these genes” is not supported (note that the association of regulatory elements to genes is based on proximity for practical analytical reasons; without a transcriptional response to TF manipulations, the association between a TF and a gene is inadequate).

Response: To support the conclusion that Myogenin, Tead4, and Klf5 positively activate the expression of muscle developmental genes, we performed loss of function experiments of Myogenin, Tead4, or Klf5 with shRNA lentivirus infection in C2C12 cells differentiated for 3 days. We observed that the expression of *Myomaker*, *Col25a1*, *Tnnt2*, and *Mymx* are significantly reduced when Myogenin, KLF5, and Tead4 expression is reduced. These results confirm our conclusion that Myogenin, Tead4, and Klf5 positively activate the expression of the muscle developmental genes and the differentiation of skeletal muscle. We included these data in revised Figure S4c.

-The muscle phenotype in the *Maf* KO mice is very interesting. Moreover, the specific deregulation of mature genes and not of genes linked to the developmental program is excellent. One aspect that is probably worth discussing is that *Maf* was identified as a TF of adult myofibers yet the KO mice have a clear muscle phenotype at E18.5. Also, it should be explained why you made a new KO mouse since one is already available. It is unfortunate that the new *Maf* allele is not Cre-inducible, which would allow spatiotemporal regulation (and study specifically the role of adult *Maf*).

Response: *Maf* and mature muscle genes (*Myh4*, *Actn3*, *Mybpc2*, *Ckm*, and *Atp2a1*) start to be expressed at low levels in fetal myofibers. After birth, the expression of the mature genes dramatically increases. *Maf* KO mice show a defect in mature gene

expression at E18.5 and muscle hypoplasia. We modified the Discussion to better explain this point (page 11 of the revised manuscript), and see below:

“Maf is not expressed in muscle progenitor cells (MuSCs and myoblasts), and it begins to be expressed and active in a subset of fetal myofibers (Cluster 5). In these cells, Maf promotes myofiber maturation by directly activating the expression of the mature gene program, as seen by the decreased expression of mature genes (*Myh4*, *Actn3*, *Mybpc2*, *Atp2a1*, and *Ckm*) and the muscle hypoplasia in E18.5 Maf KO muscles.”

We were not able to obtain the published Maf KO mice from a private company or from an academic research team. We therefore decided to generate a mouse line of KO mice for Maf. Because global knockout mice can be generated faster to generate than conditional knockout mouse, we decided to first analyze the muscle of the MAF global knockout mice. We are also generating MAF conditional knockout specifically in muscle fibers, to complete our understanding of the role of this TF in muscle fibers in a future study, but this will take at least another year.

Less major and editorial comments:

-Please, give more information on the samples that were analysed for the snRNAseq/ATACseq experiments. It is unclear in the Methods if the P5 sequencing was done on whole hindlimbs or dissected muscles.

Response: As stated previously, nuclei at all times were extracted from skeletal muscle dissected from the whole hindlimb. We provided more information on the samples that were analysed in the Results and Materials and Methods of our revised manuscript (pages 4 and 14 of the revised manuscript).

-“ The UMAP resembled a rainbow with myonuclei separated into nine different clusters”.

What is the scientific meaning of a UMAP resembling a rainbow? Is this metaphor meaningful in some way?

Response: This metaphor is used to describe the shape of the UMAP. We modified this sentence in page 5 to better explain it.

- Page 9 “The decreased expression of these mature genes reflects the lack of expression of Maf in Cacna1s KO myofibers”. Do you mean “The decreased expression of these mature genes IS REFLECTED BY the lack of expression of Maf in Cacna1s KO myofibers”?

Response: Thank you for noting this error. We updated the manuscript accordingly.

-You may want to reconsider the title as, in my view, Maf is not the main finding of this study.

Response: Thank you for your suggestion. We modified the title of our article to better fit with the main findings of our study. “Opposing gene regulatory programs governing myofiber development and maturation revealed at single nucleus resolution”.

Reviewer #2 (Remarks to the Author):

In the current manuscript, Matthieu et al., performed snRNA-seq and snATAC-seq of skeleton muscle cells undergoing maturation from perinatal to postnatal stage. During perinatal development, they demonstrate that Myogenin, Klf5, and Tead4 cooperatively activate **cardiac** genes and facilitate the maturation of myocytes. At the postnatal stage, the authors identified Maf as a novel transcription factor to activate the maturation program. They further discovered the L-type Calcium channel regulates the myocyte

maturation by repressing the activity of Myogenin, Klf5, and Tead4, and activating Maf. Overall, the dataset generated in current manuscript is a valuable resource for the understanding of skeleton muscle development. However, there are several major concerns needed to be addressed before the publication of the manuscript.

Response: Thank you for your careful review of our manuscript. We have addressed your comments and concerns as detailed below, and revised our manuscript accordingly. We would like to clarify that our study demonstrates that Myogenin, Klf5, and Tead4 activate the expression of genes required for myofibers development in skeletal muscle and not cardiomyocytes in the heart.

Major:

1. The bioinformatic analysis performed on the multi-omics dataset is quite minimal. More in-depth analyses, even using standard and established algorithms, could greatly benefit the manuscript. For example, the authors could reconstruct pseudotime trajectory or calculate RNA velocity to have a continuous alignment of cells undergoing maturation. Such analysis would allow them to explore the genes with different expression patterns along the maturation process without bias.

Response: We thank the reviewer for suggesting a more in-depth bioinformatic analysis. We performed Pseudotime analysis of our data with Monocle. We observed a similar trajectory of myonuclei differentiation with the Pseudo-time and Real-time analysis. We included these data in revised Figure S3a.

However, in adult myofibers, the Pseudotimes analysis classifies the different fibers in different pseudotime. We believe that this is a bias of this analysis and not happening in an adult skeletal muscle. Interpretation of the pseudotime analysis can be challenging and lead to mistakes:

“Interpretation of the resulting data is challenging and requires computational models at multiple levels of abstraction”

<https://journals.biologists.com/dev/article/146/12/dev170506/19458/Concepts-and-limitations-for-learning>

RNAvelocity analysis was designed for single cell RNAseq analysis and assumes that "spliced mRNA abundance (RNA velocity) is determined by the balance between the production of spliced mRNA from unspliced mRNA, and the mRNA degradation" (RNA velocity of single cells, La Manno et al., 2018) The percentage of intronic reads in single nucleus RNAseq is higher compared to single cell RNAseq, and does not depend on the balance between the production of spliced mRNA from unspliced mRNA and mRNA degradation. The limitations of RNA velocity analysis methods for single nucleus RNAseq have been discussed here:

<https://github.com/theislab/scvelo/issues/118>

<https://github.com/velocyto-team/velocyto.R/issues/97>

2. In Figure 2, it is unclear how the enriched motifs were identified from the multi-omic analysis. The methods are missing in the method section. If the authors used chromVar, please include the motif activity score alongside the expression heatmap (Figure 2e) to justify their selection of the motifs to follow up.

Response: Thank you for pointing out this oversight. Overrepresented motifs were identified using the software Chromvar. We included the motif activity score alongside the expression heatmap as suggested in Figure 2f. We added the link to the script for the analysis in the method section:

<https://github.com/matthieudossantos/Multiome-snRNAseq-and-snATACseq-Muscle-development-atlas-script>

3. In Figure 3, the overlapped peaks in only 0.2% of the total peaks, this is too low. The authors need to perform statistical analysis to test whether such overlap is significant or is merely odds (insignificant thus lacks biological meaning). Also, within the 126 peaks, are these peaks mainly associated with the myocyte maturation? GO analysis would be helpful.

Response: To show Myogenin, Klf5, and Tead4 synergistically activate the expression of developmental muscle genes, the following analyses were performed:

1. We performed fisher's exact test on the number of overlaps/unique intervals between Myogenin vs. Tead4 peaks, Myogenin vs. Klf5 peaks, as well as Klf5 vs. Tead4 peaks. In brief, we used bedtools fisher command to test if the amount of overlap between the 2 sets of intervals is more than expected given their coverage and the size of the genome.

Myogenin vs. Tead4: fisher's exact test two-tail p-value = 0

Myogenin vs. Klf5: fisher's exact test two-tail p-value = 0

Klf5 vs. Tead4: fisher's exact test two-tail p-value = 1.8816e-289

These extremely small p-values support the conclusion that their pairwise overlap is statistically significant.

2. Given that the number of Tead4 peaks (1429) is dramatically smaller than the number of Myogenin peaks (56524) using the same peak calling parameters, we reasoned that our Tead4 peak calling might miss peaks when using the same peak calling threshold as Myogenin and Klf5. To avoid the reliance on peak calling threshold choice, we set out to see whether the ChIP signals across 3 TFs are highly correlated instead. To do so, we first merged the peaks from all 3 TFs to form a reference and checked the global patterns across all three TFs against this merged reference. The global ChIP signal pattern shown

in the heatmap indicates the high correlation among the 3 TFs (see below). Furthermore, the correlation among the 3 TFs is very high (see scatter plot figure).

Myogenin Tead4 Klf5

3. We performed Gene ontology analysis on the genes associated with the 126 peaks cobound peaks by Myogenin, Klf5 and Tead4 using Metascape. We observed that these

peaks were associated with several GO terms included myoblast fusion. We did not include this figure in our manuscript to avoid overloading our article with data and information.

4. As the authors used an in vitro differentiation of C2C12 to validate the function of Myogenin, Klf5, and Tead4, it is critical to know if the binding of these TFs correlate with the open chromatin accessibility during muscle maturation from their scATAC-seq. Additionally it is recommended to provide the information if there is a change in DNA accessibility along the muscle maturation on these TFs' binding peaks.

Response: Thank you for your suggestions. We added the chromatin accessibility track from E18.5 and 2 months old myonuclei in figures 3c and 4f. We observed a strong correlation between the binding of developmental TFs (Myogenin, Tead4, and Klf5), the active histone marks, and the opening of the chromatin in developmental myonuclei. In adult myonuclei, these ATAC peaks are closed, suggesting that Myogenin, Klf5, and Tead4 do not bind this DNA region in the mature stage.

5. The last part of the manuscript seems to be disjoint from the previous section of the manuscript. A better transition or justification is needed to switch from the multi-omics data analyses to Cacna1s KO experiment.

Response: Thank you for pointing this out. We have provided more information regarding the link between electrical stimulation and myofiber maturation. We hope this modification (page 9 of our revised manuscript) will better justify the experiments on Cacna1s KO mice.

6. Along the same line, the conclusion made from the bioinformatics analysis (before Cacna1s KO part) lacks support both in silicon and experimentally. To establish the link between the Cacna1s and Maf, the authors may consider analyzing how many Mad target genes are dysregulated in Cacna1s KO muscle cells. Similarly, the slight increase in the expression of Myogenin, klf5, and Tead4 after Cacna1s KO is not sufficient to support they act downstream of Cacna1s. To support such claim, a rescue experiment is needed.

Response: Among the 278 genes significantly downregulated in Cacna1s KO muscle compared to control (p value < 0.05), 138 of them contain a Maf binding peak in their promoter or close to the promoter. We added a Venn diagram in Supp Fig. 7d to represent the number of genes downregulated in Cacna1s KO myonuclei that possesses Maf binding peak in ChIP-seq data. The expression of *Myogenin*, *Klf5* and *Tead4* increases in Cacna1s KO, but we do not think we can rescue the phenotype of Cacna1s KO by decreasing the expression of Myogenin, Klf5 and Tead4 (caused by excitation-contraction coupling defect).

Minor:

1. For Figure 2c legend, it should be top ten instead of top five marker genes in each cluster. Please double check.

Response: Thank you for pointing out this error. We corrected this in our revised legends.

2. In Extended Figure 3c, UMAP for Klf5 is missing.

Response: We added the UMAP for Klf5 to the revised manuscript Extended Figure 3c.

3. Please add the genomic coordinate to Figure 3C, 4F.

Response: We added the genomic coordinate to Figure 3C, 4F to the revised manuscript.

4. In Figure 5g, these genes are all the direct targets of Maf. Does the author also observe changes of cardiac maturation genes which are not directly bound by Maf?

Response: We performed RT-qPCR to measure the expression level of 3 mature cardiac genes (*Ryr2*, *Myh6*, *Myl2*) that are not directly bound by Maf in the skeletal muscle of WT and Maf KO mice. We observed no significant difference in the expression of these cardiac maturation genes in absence of Maf in skeletal muscle.

Reviewer #3 (Remarks to the Author):

During development, defined gene regulatory networks govern sequential steps of myogenesis, from the specification of myogenic progenitors, to differentiating myoblasts that exit the cell cycle and begin the process of fusion to generate the multinucleate

myofibre. However, the transcriptional regulators of muscle maturation remain undefined. While it is known that innervation and electrical stimulation of the myofibre promotes maturation, less is known how innervation of the myofibre modifies transcriptional activity.

In this manuscript, Dos Santos et al. present data delineating the formation and maturation of skeletal muscle based on gene expression programs revealed by combined single-nucleus RNA-sequencing and single-nucleus ATAC-sequencing. They analyzed and compared embryonic (E14.5), fetal (E18.5), neonatal (P5), and adult (2 months) hindlimb skeletal muscle. The authors propose a model of developmentally regulated muscle genes by the synergistic activity of the transcription factors Myogenin, Klf5, and Tead4. They further show that the mature myofiber gene expression program is controlled by the transcription factor Maf. Maf knockout mice die perinatally and display signs of disturbed skeletal muscle development. Moreover, the authors use a *Cacna1s* knockout mouse model, which is characterized by paralyzed and atrophic muscle, an immature myofiber transcriptome, to demonstrate that Maf expression is reduced upon defective upstream motor innervation.

There are many strengths in this manuscript. Overall, the manuscript presents high quality data obtained from state-of-the-art nucleotide sequencing. The single nuclear RNA-sequencing and single nuclear ATAC-sequencing datasets will be a significant contribution to the myogenesis community. Moreover, the subsequent demonstration of how these snRNA-seq and snATAC-seq datasets can be analyzed to reveal novel regulators of the myogenic program (Myogenin-Klf5-Tead4 \diamond Maf) is excellent. The use of mouse genetic tools provides, in my opinion, the cleanest demonstration of the importance of Maf for myofibre maturation. Overall, the authors provide an interesting model by which the Ca²⁺ transport activity of *Cacna1s* promotes Maf-dependent myofibre maturation, while repressing a Myogenin-Klf5-Tead4 regulatory network that is predominant in muscle development.

Response: Thank you for your enthusiasm about our paper and the valuable comments.

The authors interesting model can be better supported by addressing weaknesses in the manuscript as follows:

Major points (revisions needed)

1.The authors rely heavily on their snRNA-seq and snATAC-seq datasets to demonstrate coexpression of Myogenin with Klf5 and Tead4. The work would be better supported by protein analyses, for example immunofluorescence for these TFs in fetal vs adult myofibres.

Response: To demonstrate the coexpression of Myogenin with Klf5 and Tead4 in developmental myofibers and not adult myofibers, we performed Western Blot to measure their protein expression levels in hindlimb skeletal muscle at E14.5, E18.5, P5, and adult. We observed that all three TFs are highly expressed in E14.5 and their expression gradually decreased during development. In adult muscles, these 3 TFs are either undetectable or minimally expressed. We added these data in Supplemental Figure 4a.

2.The authors provide some evidence that Myogenin-Tead4-Klf5 transcription factors cooperate to regulate gene expression. In addition to these TFs co-occupying peaks, the authors provide some evidence of interaction in 293 cells. Synergistic transcriptional activity is demonstrated with a myomaker enhancer-promoter luciferase assay in 293 cells. While the presented data is strong and should be included, there is a lack of data supporting the interactions in myogenic cells. These interactions are potentially revealed in fetal myoblasts or at the correct myogenic stages using proximity ligation assay (or similar strategy) to reveal interactions in situ and/or immunoprecipitation of endogenous complexes from myoblasts. As the myofibre matures, one might expect loss of these interactions, which can be reported with the same tools. If feasible, investigating these interactions in the Maf KO and Cacna1s KO would also support Figure 5, Figure 6 and the authors presented model (see also point 4 below).

Response: To confirm the interactions of Myogenin, Klf5, and Tead4 in myogenic cells, we performed Co-immunoprecipitation of endogenous complexes from C2C12 cells differentiated for 2 days. We observed by Co-IP that endogenous Myogenin can interact with Klf5 and Tead4. This result confirmed the endogenous interaction of these TFs in myogenic cells. We added these data in Supplemental Figure 4e.

3. The use of mouse genetic tools in the manuscript is a strength of the manuscript, but the solitary use of H&E staining for analysis, nor any quantification of the analyses, does not lead to high quality data. Moreover, the H&E data presented do not convincingly demonstrate a muscle phenotype. The analyses would be improved for example using immunofluorescence with antibodies against Myogenin, embryonic myosin heavy chain, perhaps Myh3 and fast Myh8 (see for example Figure 4B, slow vs fast muscle Maf expression levels). If antibodies are available, immunofluorescence or immunoblotting analyses for genes shown by the authors to be regulated by MAF (eg/ Myh4, Actn3) would support the authors conclusions that MAF regulates myofibre maturation.

Response: The histological analysis of hindlimbs by H&E staining revealed a significant 32% reduction of the tibialis anterior area in Maf KO mice compared to WT, as presented in Figure 5c and quantified in Figure 5d. To better observe the decreased in size of the KO Maf fibers, we took better zoom images of the fibers in hematoxylin and eosin sections. Transverse and longitudinal sections showed that Maf KO myofibers were multinucleated but appeared smaller compared to WT. Immunofluorescence with antibodies against embryonic myosin heavy chain Myh3 and neonatal Myh8 and slow Myh7 are presented in Supp5e. We did not observe a major difference in expression of the developmental (Myh3 and Myh8) and slow Myh (Myh7) genes. We performed western blot against Actn3 and Mybpc2 in E18.5 WT and Maf KO hindlimb skeletal muscle.

We did not observe a significant changes of Actn3 and Mybpc2 proteins in Maf KO compared to WT. These expression differences between mRNA and protein could be due to a difference in sensitivity of western blot and RT-qPCR, or a biological difference between mRNA abundance and protein. Our conclusions that Maf regulates the expression of mature fast muscle genes (*Myh4*, *Actn3*, *Mybpc2*, and *Atp2a1*) has been confirmed by a recent article (Large Maf transcription factor family is a major regulator of fast type IIb myofiber determination, Sadaki et al., 2023).

4. Figure 6, the model would be better supported if the authors show *Cacna1s* KO embryos have reduced MAF protein expression, potentially shown by immunoblotting as done in Figure 5A or by immunofluorescence analyses if the antibody is suitable. Likewise, increased myogenin KLF5, TEAD4 protein expression could be shown using immunoanalyses (see also the end of point 2 above). I understand the authors present high quality nucleotide sequencing data for these genes, but in my opinion these datasets are starting points for deeper investigation.

Response: We performed western blot using protein extracted from E18.5 hindlimb skeletal muscle of WT and *Cacna1s* KO mice. We observed a strong decrease of Maf expression in the KO muscle compared to WT, and increased expression of Myogenin, Klf5, Tead4 proteins. We added these data in Supplemental Figure 7f.

5. There is no mechanistic insight into how Ca²⁺ influx from CACNA1S regulates Maf and/or Myogenin-Klf5-Tea4 expression. If known Ca²⁺ responsive regulators such as NFAT or CaMK pathways are not involved, I understand that this point may require deeper investigation outside the context of experimental revisions.

Response: The mechanism whereby of how Ca²⁺ influx regulates TF gene expression has been previously studied (reference below). These studies showed that electrical activity represses Myogenin gene expression and Myogenin transcriptional activity via CaMKII phosphorylation:

Merlie, J.P., Mudd, J., Cheng, T.C. & Olson, E.N. Myogenin and acetylcholine receptor alpha gene promoters mediate transcriptional regulation in response to motor innervation. *J Biol Chem* **269**, 2461-2467 (1994).

Tang, H. *et al.* CaM kinase II-dependent phosphorylation of myogenin contributes to activity-dependent suppression of nAChR gene expression in developing rat myotubes. *Cell Signal* **16**, 551-563 (2004).

We do not know the mechanism regulating the expression of Tead4, Klf5, and Maf via calcium and muscle contraction. Several signaling pathways candidates could be involved. The signaling pathways regulating the expression of these TFs in developing, mature and disease myofibers could be the subject of future studies.

6. Figure 4b needs to be presented with error bars, sample size and statistics.

Response: We now performed the RT-qPCR analysis in multiple tissues using n=3 mice. We have updated the error bars, sample size and the statistical comparison between the Gastrocnemius and the intestine.

Minor points (do not necessarily require experimental revisions)

1. Figure 4B, the authors reveal increased Maf mRNA expression in fast vs slow muscle groups, but do not comment. If these changes are significant, is MAF more important for transcription of fast myosin heavy chains vs slow?

Response: Maf is more highly expressed at the mRNA level in myofibers expressing the fast myosin heavy chains 4 (Myh4) than the slower isoforms Myh1, Myh2 and Myh7 (Dos Santos et al., 2020). In single nucleus ATACseq data, the Maf binding motif is more active in fast Myh4 myofibers than in Myh1, Myh2, and Myh7 fibers (Dos Santos et al., 2020). Western blot data from adult skeletal muscle revealed a higher accumulation of Maf mRNA and protein in fast skeletal muscle fibers compared to slow (Fig. 4c). The ChIP-

seq of Maf in adult skeletal muscle revealed specific binding of Maf in the Myh4 promoter and not in the Myh1, Myh2, and Myh7 promoters (Extended Data Fig.5A). Maf is more important for transcription of fast myosin heavy chains compared to slow myosin heavy chains. An article published during the revision of our manuscript supports the same conclusions (Large Maf transcription factor family is a major regulator of fast type IIb myofiber determination, Sadaki et al., 2023). We added a comment on the role of Maf in fast myofiber determination in the discussion (page 12 of our revised manuscript).

2.The authors provide evidence that MAF is required for maturation of the myofibre, but do not comment on whether MAF is needed for maintenance of the myofibre. It would be interesting to know if Maf/MAF expression is decreased upon denervation of adult muscle, or perhaps on single EDL myofibres that have been isolated from the motor neuron, fresh isolated or cultured for short periods of time.

Response: We agree with the reviewer that it would be interesting to study whether MAF is needed for maintenance of myofiber during denervation. However, we believe that this is beyond the scope of our current study, which focuses on characterizing the gene regulatory networks that control myofiber development and maturation. We are pursuing this direction to study the role of Maf during muscle atrophy in a separate study.

3.Figure 1. Was the identified MuSC cluster unique to adult muscle? I initially had some concerns with respect to labeling myogenic progenitors present at developmental stages as MuSCs. In Figure 1A, should the MuSC be labeled at 2 months with the mature myofibres? Some clarification would help.

Response: MuSCs were observed at all stages. Below is a UMAP plot showing a zoom of the MuSC cluster with the time origin of each nucleus on the right. The MuSC cluster is not unique to adult muscle, however, we observed a separation between developmental and adult MuSC in the UMAP, reflecting their transcriptomic differences. We believe that this is beyond the scope of our current study and we did not further investigate this issue.

snRNA-seq

Developmental stage

15

10

REVIEWERS' COMMENTS

Reviewer #1 (Remarks to the Author):

The revised manuscript by Dos Santos et al. demonstrates significant improvements, with the authors effectively addressing most of the reviewers' comments. This study holds great interest for both developmental biologists and muscle experts alike. The topic is expansive, requiring the application of various whole-genome protocols. Given the project's high throughput nature, I suggested validating some intriguing candidate genes on muscle sections through FISH experiments. The authors conducted FISH experiments for Col22a1 and Col25a1, confirming their spatiotemporal patterns. However, these genes were not the best choice, as their expression patterns were already known. Additionally, Col22a1 is known to be strongly expressed also in adult MTJ (eg. see Myoatlas). It would have been preferable to select novel age-specific DEGs identified through their analysis and investigate their spatial and temporal localization. The same criticism applies to the FISH experiments involving Myomaker, Myomixer, Maf, and Myh4, as the gene selection was suboptimal, except for Maf, which is further analyzed later in the study. It is somewhat expected that Myomaker and Myomixer would not be present in resting adult muscles since fusion does not occur under those conditions. It is unfortunate that the authors did not choose from the numerous novel age-specific DEGs identified through their analysis.

The revised manuscript includes a set of new experiments that effectively illustrate the formation of a trimeric transcriptional complex involving Myogenin, Klf5, and Tead4. This complex works synergistically to activate the expression of muscle genes in developing myofibers. However, there is a noteworthy comment regarding Extended Data Fig. 4f, where the presence of KLF5 in the Myogenin pull-down experiment is unclear. To ensure the accuracy of the interaction, it is recommended that this experiment be repeated for verification purposes.

Despite having some weaknesses, the manuscript overall serves as a valuable resource for enhancing our understanding of skeletal muscle development. The authors have made substantial improvements and addressed most of the reviewers' comments, contributing to the credibility and reliability of the findings. While there may be some issues with gene selection and experimental clarity in certain sections, the study's comprehensive exploration of different whole-genome protocols, the demonstration of trimeric transcriptional complex, and the role of Maf as an essential regulator of myofiber maturation highlight its significant contributions and adds valuable insights to the field. Therefore, despite its limitations, this manuscript remains a valuable asset for researchers seeking to deepen their knowledge of skeletal muscle development.

Reviewer #2 (Remarks to the Author):

The authors have satisfactorily addressed all of my concerns. In particular, some of the confusions or questions related to their single nucleus RNA-seq and ATAC-seq data have been clarified with explanation and/or additional new information. The manuscript has been greatly improved, and is acceptable for publication.

Reviewer #3 (Remarks to the Author):

The authors have satisfactorily addressed my review of the manuscript. Overall, it is an interesting study that initially outlines the developmental course of myonuclei maturation,

which gave rise to more in depth analyses of opposing gene regulatory programs governing myofiber development (Myogenin-Klf5-Tead4) and maturation (Maf-dependent).

REVIEWER COMMENTS

Reviewer #1 (Remarks to the Author):

The revised manuscript by Dos Santos et al. demonstrates significant improvements, with the authors effectively addressing most of the reviewers' comments. This study holds great interest for both developmental biologists and muscle experts alike. The topic is expansive, requiring the application of various whole-genome protocols. Given the project's high throughput nature, I suggested validating some intriguing candidate genes on muscle sections through FISH experiments. The authors conducted FISH experiments for *Col22a1* and *Col25a1*, confirming their spatiotemporal patterns. However, these genes were not the best choice, as their expression patterns were already known. Additionally, *Col22a1* is known to be strongly expressed also in adult MTJ (eg. see Myoatlas). It would have been preferable to select novel age-specific DEGs identified through their analysis and investigate their spatial and temporal localization. The same criticism applies to the FISH experiments involving *Myomaker*, *Myomixer*, *Maf*, and *Myh4*, as the gene selection was suboptimal, except for *Maf*, which is further analyzed later in the study. It is somewhat expected that *Myomaker* and *Myomixer* would not be present in resting adult muscles since fusion does not occur under those conditions. It is unfortunate that the authors did not choose from the numerous novel age-specific DEGs identified through their analysis.

Response: Thank you for your insightful comments. We have addressed your comments and concerns as detailed below and revised our manuscript accordingly. We chose the genes *Col22a1*, *Col25a1*, *Myomaker*, *Myomixer*, *Maf*, and *Myh4* because their functions and expression patterns were partially known, thereby enabling us to better understand the function and location of the different clusters in the snRNAseq data. With regard to novel differentially expressed genes (DEGs), we performed new Fish experiments with two genes with unknown functions in skeletal muscle, *Fstl4* and *Tmem108*, which are specifically expressed in developing myofibers (Extended Data Fig. 3d).

The revised manuscript includes a set of new experiments that effectively illustrate the formation of a trimeric transcriptional complex involving Myogenin, Klf5, and Tead4. This complex works synergistically to activate the expression of muscle genes in developing myofibers. However, there is a noteworthy comment regarding Extended Data Fig. 4f, where the presence of KLF5 in the Myogenin pull-down experiment is unclear. To ensure the accuracy of the interaction, it is recommended that this experiment be repeated for verification purposes.

Response: To verify the formation of the transcriptional complex of Myogenin, Tead4, and Klf5, we performed new Co-IP experiments of endogenous Tead4 and its interaction with Klf5. The results are more clear than the previous Co-IP and confirm the formation of the transcriptional complex between Myogenin, Tead4, and Klf5. We decided to replace the previous Co-IP of Myogenin and Klf5 with the new endogenous Co-IP of Tead4 and the interaction of Klf5 (Extended Data Fig. 4f). The original uncropped blots are in the source data file.

Despite having some weaknesses, the manuscript overall serves as a valuable resource for enhancing our understanding of skeletal muscle development. The authors have made substantial improvements and addressed most of the reviewers' comments, contributing to the credibility and reliability of the findings. While there may be some issues with gene selection and experimental clarity in certain sections, the study's

comprehensive exploration of different whole-genome protocols, the demonstration of trimeric transcriptional complex, and the role of Maf as an essential regulator of myofiber maturation highlight its significant contributions and adds valuable insights to the field. Therefore, despite its limitations, this manuscript remains a valuable asset for researchers seeking to deepen their knowledge of skeletal muscle development.

Reviewer #2 (Remarks to the Author):

The authors have satisfactorily addressed all of my concerns. In particular, some of the confusions or questions related to their single nucleus RNA-seq and ATAC-seq data have been clarified with explanation and/or additional new information. The manuscript has been greatly improved, and is acceptable for publication.

Response: Thank you for your careful review of our manuscript and for your enthusiasm.

Reviewer #3 (Remarks to the Author):

The authors have satisfactorily addressed my review of the manuscript. Overall, it is an interesting study that initially outlines the developmental course of myonuclei maturation, which gave rise to more in depth analyses of opposing gene regulatory programs governing myofiber development (Myogenin-Klf5-Tead4) and maturation (Maf-dependent).

Response: Thank you for your enthusiasm about our paper and the valuable comments.